# Mammal responses to human recreation depend on landscape context

Solène Marion [1,2]*, Gonçalo Curveira Santos[1,3,4], Emily Herdman[5], Anne Hubbs[6], Sean Patrick Kearney[1], A. Cole Burton [1,7]*

**1** Department of Forest Resources Management, University of British Columbia, Vancouver, British Columbia, Canada, **2** JNCC, Inverdee House, Aberdeen, United Kingdom, **3** CIBIO—Research Center in Biodiversity and Genetic Resources, InBIO Associated Laboratory, Universidade do Porto, Vairão, Portugal, **4** BIOPOLIS Program in Genomics, Biodiversity and Land Planning, CIBIO, Vairão, Portugal, **5** Innotech Alberta, Edmonton, Alberta, Canada, **6** Resource Stewardship, Environment and Protected Areas, Alberta, Canada, **7** Biodiversity Research Centre, University of British Columbia, Vancouver, British Columbia, Canada

* solene-marion@orange.fr (SM); cole.burton@ubc.ca (ACB)

## Abstract

Rapid growth in outdoor recreation may have important and varied effects on terrestrial mammal communities. Few studies have investigated factors influencing variation in observed responses of multiple mammal species to recreation. We used data from 155 camera traps, in western Alberta (Canada), and a hierarchical Bayesian community modelling framework to document 15 mammal species responses to recreation, test for differential responses between predators and prey, and evaluate the influence of local context. Factors characterizing context were trail designation (i.e., use by motorized vs non-motorized), management type, forest cover, landscape disturbance, and season. We used three measures to characterize variation in recreation pressure: distance to trail, trail density, and an index of recreation intensity derived from the platform Strava. We found limited evidence for strong or consistent effects of recreation on mammal space use. However, mammal space use was better explained by an interaction between recreation and the influencing factors than by either on their own. The strongest interaction was between trail density and management type; mammals were more likely to avoid sites near a higher density of trails in areas with more restrictive management. We found that responses to recreation varied with the trail designation, although there were not clear or consistent differences between responses to trails designated for motorized vs. non-motorized use. Overall, we found that responses were species- and context-dependent. Limiting the density of trails may be important for reducing negative impacts to mammals within conservation areas. We show that using multiple measures of recreation yields more insight into the varied effects of human disturbances on wildlife. We recommend investigating how different characteristics of recreation (noise, speed, and visibility) influence animal behaviors. Multispecies monitoring and modelling across multiple landscapes that vary in recreation pressure can lead to an adaptive management approach to ensuring outdoor recreation coexistence with wildlife.

**Data Availability Statement:** All files are available from the GitHub page:: https://github.com/solenemarion/multispecies_model. Data can be accessed here: Marion, Solene; (2023). Mammal

responses to human recreation depend on landscape context. figshare. Dataset. https://doi.org/10.6084/m9.figshare.24418257.v1.

**Funding:** The Wildlife CAMERA project was funded by Alberta Innovates, Innotech Alberta, and Alberta Environment & Parks (now Alberta Environment and Protected Areas). Innotech Alberta was a collaborating partner and funder of this study. Specifically, staff of Innotech Alberta helped design the camera trap survey and collect field data, and Innotech Alberta provided funding to support data analysis and manuscript preparation. E.H. provided scientific feedback during manuscript preparation as part of the co-authorship team. Other funders had no role in study design, data collection and analysis, decision to publish, or preparation of the manuscript. Additional support for this research was received from the University of British Columbia and the Natural Sciences and Engineering Research Council of Canada (Canada Research Chair and Discovery Grant RGPIN-2018–03958 to A.C.B).

**Competing interests:** The authors have declared that no competing interests exist.

## Introduction

National parks, protected areas, and other natural areas are the cornerstone of global biodiversity conservation efforts, but are increasingly attracting visitors [1, 2]. These areas are particularly attractive for nature-based tourism and outdoor recreation such as hiking, mountain biking, or motorized activities (e.g., all-terrain vehicles, snowmobiles) [3]. While such recreation activities can generate economic and health benefits, they can also affect various components of the environment, including terrestrial wildlife [4, 5]. Recreation activities can impact wildlife in multiple ways, including positive or negative effects on animal use of space, such as facilitated movement along trails [6] or displacement from important habitats [7]. The considerable variation in observed effects of recreation on wildlife has been attributed to multiple factors that can influence animal responses to outdoor activities (hereafter "influencing factors") [8]. Tablado & Jenni (2017) [8] classified these factors into three categories (i) recreation characteristics (ii) spatio-temporal context (iii) intrinsic characteristics of the animals.

Influencing factors such as the location, intensity, and type of recreation can affect animal responses. For example, elk (*Cervus elaphus*) in North America were more sensitive to off-road than on-road all-terrain vehicles [9]. Similarly, in the United States, carnivore species were more sensitive to recreation (e.g., hiking and camping) in protected areas with higher levels of landscape disturbance than they were to recreation in protected areas with lower levels of disturbance [10]. Moreover, depending on the species, different types of recreation have also been found to have different impacts on wildlife, such as stronger effects from motorized vehicles and mountain biking than hiking or horseback riding, potentially due to elevated levels of noise and speed [11, 12]. Mammals perceive recreationists visually, auditorily, and olfactorily [8] and this perception can be modified by such varied factors as recreationists' color of clothing (but see [13]) or speed of movement [8].

Animal perceptions and responses to recreation may also be influenced by the spatio-temporal context within which interactions occur. For instance, high-density vegetation can reduce noise pollution and direct lines of sight, and thus habitats with lower visibility or sound may be preferred by wildlife in recreational landscapes [14]. Similarly, the same species may also show different responses to recreation in different areas or seasons [15, 16]. For example, Barja *et al*. 2007 [17] found that European pine martens (*Martes martes*) were most sensitive to human and other environmental disturbances during the reproductive season. Mammals can also habituate to human presence and infrastructure, and accordingly display fewer responses to recreation in busier or more developed areas. For instance, Prices *et al*. (2014) [18] found that responses of mule deer (*Odocoileus heminonus*) to humans were reduced near a biological field station in Colorado. Habituation to disturbances is a common result of outdoor recreationists-wildlife interactions [19, 20]. It depends on the animal's risk perception and time exposition to the disturbance: the more the animal is exposed to human the more it is habituated [21]. Habituation and risk perception can be altered if hunting occurs simultaneously with outdoor recreation, leading to challenges in disentangling the various impacts of human disturbance [22–24].

Similarly, variation in the ecological and life-history traits of mammal species can also influence their perceptions and tolerances of risk from human disturbance [8]. For example, mammal responses to stress differ depending on their trophic level. Prey display vigilance behaviour, or aggregate in large groups, in response to stress [24, 25], while such behaviours are less often observed in predators. Specifically, the difference between predator and prey responses to recreation has been increasingly investigated in the context of the "human-shield hypothesis" [26–29]. This hypothesis suggests that predators are more sensitive to, and thus avoid, people, while prey find refuge from predators by associating with human activities.

Underlying mechanisms of such hypotheses are still unclear; however, they make clear the importance of understanding variation in responses to recreation among species and guilds.

Previous studies of recreation impacts on wildlife have often focused on the relationship between one focal species and one type of recreation (e.g., elk and skiers [30]; red squirrels and mountain bikers [31]). While this approach can inform single-species conservation or the management of one activity, it may be inadequate to capture broader community dynamics and the complexity of the cumulative effects of multiple activities on wildlife [32]. For example, Procko *et al.* (2023) [33] found that human presence had different impact on wildlife nocturnality depending on the species. Similarly, human disturbance can lead to cascading effects that can alter community structure and ecosystem functioning such as inter-species interactions [34, 35]. Long-term exposure to human disturbance can impact animal fitness (i.e., survival and reproduction success) [36, 37]. However, different outcomes of human disturbance impacting the long-term can be found. Salvatori *et al.* (2023) [37] found that both community and species-level occurrences increased within a protected area in Europe, despite human activities causing strong temporal avoidance in the whole community. This highlights the complexity of human-wildlife interaction and the necessity of long-term studies. Lastly, alterations in herbivory and predation resulting from recreational activities can have broader environmental implications through functional ecosystem change, as demonstrated in Di Nicola *et al.* (2023), where human-mediated changes were found to impact local carbon stocks [38].

Just as wildlife responses to recreation vary across species and contexts, approaches to managing the impacts of recreation vary considerably. Different types of recreation management have been used to limit the intensity of recreation disturbance, such as by limiting visitor numbers [39], or restricting the timing or location of certain recreation activities [40]. However, assessments of the effectiveness of such management actions across wildlife communities are often lacking. Thus, there is a need to evaluate how multiple species interact with different types of activities in different contexts to inform land and wildlife managers about strategies to promote human-wildlife coexistence.

Recent advances in sampling and statistical methodologies have created an opportunity to pursue community-level studies of the impacts of diverse recreation activities on wildlife. In particular, the development and popularization of motion-triggered cameras, hereafter camera traps [41–43], has allowed studies that focus on wildlife responses to recreation to capitalize on the wealth of multi-species information that is collected non-invasively [44]. Similarly, the development of joint species distribution models [45, 46], such as the Hierarchical Modelling of Species Communities approach (HMSC), has opened new doors for community ecology by modelling the abiotic environment at the community level while accounting statistically for co-occurrence among species [47, 48].

In this study, we combined camera trapping and community modelling to understand how the distribution and intensity of recreation activities influences space use by multiple terrestrial mammal species and determine what are the factors influencing mammal responses. Specifically, we used multispecies modelling of camera trap detections of small-, medium- and large-bodied mammals in two areas increasingly used by recreationists in western Alberta, Canada. We hypothesized that (1) the impacts of recreation will differ among mammalian predators and prey, and be influenced by the following factors: (2) trail designation, (3) habitat structure, (4) human footprint, (5) type of management, and (6) season. We made several predictions related to these factors. First, under the human shield hypothesis, we predicted that predator species will more strongly avoid recreationists, while prey species will be attracted to areas with more recreationists and fewer predators. Second, we predicted that trails designated for motorized activities will be more strongly avoided by mammals than trails designated for non-motorized activities, due to the elevated noise and speed of motorized activities. Third, we

predicted that habitat characteristics can influence interactions between recreation and mammals, such that greater vegetation density (measured as percentage of forest) will provide more security and thereby reduce mammal negative responses to recreation. Fourth, we predicted that the percentage of human footprint on the landscape, such as roads and other infrastructure, can lead to greater habituation to people by mammals, and thus reduced responses to recreation. Fifth, we predicted that impacts of recreation will be reduced in areas where there are greater restrictions on recreation activity (e.g., reduced access). Sixth, and finally, we predicted that responses may differ between summer and winter since recreation activities and mammal ecology vary seasonally.

## Methods

We evaluated how mammal space use changed in response to recreation and influencing factors based on camera trap detections and characterizations of recreation intensity and landscape conditions around camera stations, using the HMSC multi-species modelling approach.

### Study areas

Our study was carried out across two recreational landscapes in western Alberta, Canada: Bighorn Backcountry (5,000 km$^2$) and Castle Provincial and Wildland Provincial (1,050 km$^2$) parks (hereafter Bighorn and Castle, respectively; Fig 1). Both areas have experienced a high level of recreation pressure over recent years due to their proximity to some of the most popular national parks in Canada (e.g., Banff, Jasper, and Waterton). There are more than 900 and 500 km of designated (official) trails in Bighorn and Castle, respectively, with similar trail densities in both (see Table 1). Summer recreation activities include hiking, camping, off-highway vehicle (OHV) use, fishing, hunting, cycling, and horseback riding, while winter recreation includes backcountry skiing, snowshoeing, and snowmobiling (https://open.alberta.ca, https://www.albertaparks.ca). Visitor numbers to these areas are not available; however, based on indices of recreation from the apps Strava and AllTrails (mostly used by runners, hikers, and cyclists), the level of recreation was higher in the camera trap sampling area within Castle relative to Bighorn (section 2.2; Table 1).

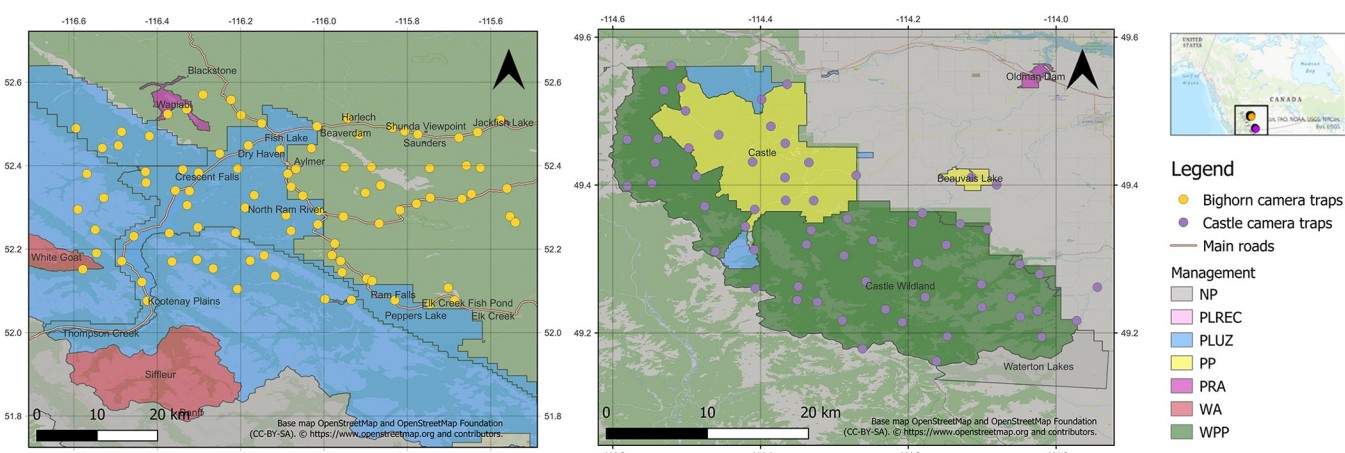

**Fig 1.** Camera traps deployed in Bighorn (left, yellow points) and Castle (right, purple points) areas in Alberta, Canada (inset). Shading represents different types of recreation management: PLREC: Public Land Recreation, PLUZ: Public Land Use Zone, PP: Provincial Park, PRA: Provincial Recreation Area, WA: Wilderness Area, and WPP: Wildland Provincial Park. Base Map: OpenStreetMap: https://www.openstreetmap.org/copyright].

**Table 1. Predictor variables used in our HMSC model to estimate effects of recreation and landscape on mammal space use.** For continuous variables, we show the mean and range (minimum and maximum) of values estimated at the camera trap sites. For categorical variables, we show the number of categories and which category was used as a reference level using an asterisk (*). BH referred to Bighorn and CA to Castle. Sources of each dataset can be found in S1 File.

| Variables | Rationale | No. categories or mean value (min, max) |
|---|---|---|
| **Recreation variables** | | |
| Distance from trail (m) | We selected distance to trail [23] and density of trails [54] as measures of recreation spatial distribution and AllTrails and Strava as measures of recreation intensity [55]. AllTrails was removed from analysis due to collinearity (see main text). | BH: 1973 (0.4, 13797), CA: 571 (1.2, 3853) |
| Trail Density (m/m² in 500m buffer) | | BH: 0.003 (0, 0.04), CA: 0.01 (0, 0.23) |
| AllTrails intensity (# reviews in 500m buffer) | | BH: 11 (0, 379), CA: 33 (0, 598) |
| Strava intensity (use rate in 500m buffer) | | BH: 5 (0, 65), CA: 36 (0, 142) |
| **Factors influencing mammal responses to recreation** (context-dependency) | | |
| Trail designation (500m buffer) | Trails can be used for different type of recreation that may affect mammals differently [12]; we predicted more negative responses to motorized activities. | 3 categories: No activities*, non-motorized, motorized. |
| Sampling Area | Mammal responses may differ between populations in different areas [15]. However, we removed this variable from further analysis due to its correlation with management type (see main text). | 2 categories (BH* and CA) |
| % Forest (500m buffer) | We used the percentage of forest as index of Habitat structure can modulate the impact of recreation on wildlife due to changes in visibility and noise attenuation [8]; we predicted less responses at sites with more forest. | BH: 76 (23, 100) CA: 60 (0, 100) |
| % human footprint (500m buffer) | Mammals can habituate to human presence [18]; we predicted less responses at sites with more footprint. | BH: 12 (0, 80) CA: 21 (0, 93) |
| Type of management | Impacts of recreation on mammals might be limited by more management restrictions on recreation activities: we predicted less impacts in areas with a greater focus on protection. | 4 levels: 1 = public lands, 2 = Public Land Use Zones (PLUZ), PLREC Public Land Recreation (PLREC), Provincial Recreation Areas (PRAs) 3 = Provincial Park (PP) 4 = Wildland Provincial Park (WPP) |
| Season | Responses may differ between summer and winter since recreation activities and mammal ecology vary seasonally. We predict that species will be differently impacted by recreation in summer and in the winter | 2 categories: Winter* (Dec 1—Apr 30), Summer (May 1—Nov 30)' which aligns with seasonal trail restriction for motorized activities in Bighorn. |
| **Additive landscape variables**: We predicted that these factors could influence mammal space use independently of recreation | | |
| Land cover | Mammals may preferentially use certain land cover (habitat types) | 4 categories: Developed (e.g., roads, power line)*, Forest, Grassland-Shrubland, Other |
| NDVI (mean per season inside 500m buffer) | Mammals may use habitats with different vegetation productivity | BH: 0.50 (0.19, 0.68) CA: 0.45 (0.22, 0.66) |
| Elevation (m) | Mammals have different elevation preferences. | BH: 1530 (1154, 2235) CA: 1595 (1341, 2178) |
| Distance to water | Water sources can be an important driver of mammal space use | BH: 96 (3, 787) CA: 132 (3, 852) |

The prevailing vegetation composition is similar in the two areas and dominated by coniferous forest. Elevation is also similar and ranges from 1000 to 3000 m, with higher, rockier alpine terrain in the west transitioning through foothills to lower forest, grassland and shrubland in the east. The region has mild summers (20 to 25˚C) and cold winters (-5 to -15˚C). Annual precipitation in the foothills of the Rocky Mountains in Alberta can reach 600 mm/year.

## Camera trap sampling

Camera trap surveys were implemented by Innotech Alberta to sample terrestrial mammals in both areas under Research and Collection Permits issued by the Government of Alberta. In Bighorn, 91 camera traps sites (single ReconyxⓇ PC900 or ReconyxⓇ HP2X camera that have similar specs) were deployed from October 2019 to August 2021, and in Castle, 64 sites were deployed from September/October 2020 to August 2022. A systematic design was used in both areas, with one camera placed in each cell of a 6 km by 6 km grid (Fig 1). Cameras were placed as close as possible to the center of each cell with a minimum distance of 2 km between camera trap sites. Cameras were not set along human (recreation) trails; thus we used different indirect proxies to recreation of activity that we describe in 2.3.1. To maximize mammal detectability, cameras were deployed facing perpendicular to game trails or natural travel corridors (e.g., drainages) near the pre-determined point, at a height of approximately 1 m off the ground and a distance of 3–5 m from the target area of expected animal movement. Three tablespoons of scent lure (O'Gorman's Long Distance Call) were added to the lower portions of a tree in the target area at about 4–7 m from the camera. Lure was used as part of broader monitoring objectives associated with this camera array, including detection of elusive species such as wolverines and fishers, which are particularly sensitive to lures [49]. However, due to few detections of these species, we did not include them in our analysis (see S1 Table), and previous research indicates that this scent lure does not significantly impact detections of other species [49]. Moreover, as lures were applied consistently at every camera trap, they didn't compromise inference about the factors that we hypothesized to influence spatial heterogeneity in species detections. Cameras were set to take one photo per trigger, with no delay between triggers, and one timelapse photo at noon every day (to ensure functionality). Cameras were visited once per year to retrieve photos and refresh batteries and lure. More frequent visits were not possible due to access limitations in the remote locations. We considered camera trap failures by examining the first and last days of daily timelapse photos, noting any days in which the camera was not functioning. We calculated the number of active sampling days for each camera trap and season to account for variation in sampling effort (see below). As a result, one camera trap with high failure rates was excluded from our analysis.

We used spatial variation in counts of camera trap detections as a measure of space use by mammal species [48]. Count data provides crucial ecological insights into how animals navigate the landscape relative to their surroundings [50]. We chose not to adopt an occupancy model framework, commonly suggested to address imperfect detection (e.g., [51]), as we believed that model assumptions would be violated (e.g., site closure, all variation in detectability modelled by covariates), introducing bias into occupancy estimates across species with varied movement behaviour [52]. We also contend that the spatial and temporal variation in frequency of detections reflects the intensity of site use rather than merely observation error [52]. For each camera trap site, we manually tallied a count of independent detections for each season, summed across the full period of sampling in each area. Site-by-season species count data were input as response variables into our multi-species model (see section 2.4). To avoid inflated counts due to repeated observations of the same individual(s) in a given detection

event, we only considered consecutive detections of the same species at the same camera to be independent if they were separated by a minimum threshold of 30 minutes [53]. We assessed animal detections separately for winter (1 December—30 April) and summer (1 May– 30 November) seasons since recreation activities (including permitted trail uses, see section 2.3.2) and animal activities vary by season. We removed detections where animals could not be identified to species level. To ensure sufficient samples of detections for modelling, we focused on 15 mammal species which each had more than 60 independent detections across Bighorn and Castle (S1 Table). Predators included grey wolf (*Canis lupus*), cougar (*Puma concolor*), coyote (*Canis lupus*), Canada lynx (*Lynx canadensis*), black bear (*Ursus americanus*), grizzly bear (*Ursus arctos*), red fox (*Vulpes vulpes*). Ungulates included elk (*Cervus canadensis*), moose (*Alces alces*), white-tailed deer (*Odocoileus virginianus*), mule deer (*Odocoileus hemionus*). Small mammals included snowshoe hare (*Lepus americanus*), marten (*Martes americana*), red squirrel (*Tamiasciurus hudsonicus*), and Columbian ground squirrel (*Spermophilus columbianus*).

### Hypothesized predictors of mammal space use

**Distribution and intensity of recreation activities.** Several different methods of directly quantifying recreation pressure have been attempted in the literature, such as the number and type of human activities detected by camera trap photos [12], or trail counters [35]. However, such methods can be labour intensive and require camera traps or counters to be set on recreation trails to obtain accurate measures, and thus risk potential tampering or concerns about privacy [56]. Since our camera traps were not set on recreation trails, we extracted four complementary variables as indirect estimates of recreation intensity at the camera site scale (Table 1). We believed these variables encompass various types of recreational activities and proxies for their spatial distribution and intensity in our study areas, in the absence of more direct observations of recreation. By incorporating them in a single analysis, we aimed to provide an holistic representation of recreationists' space use and wildlife perception/response pathways. We expected that potential disturbance to wildlife from recreation would be related to the proximity and density of recreation trails; we therefore calculated the distance from each camera site to the nearest trail [23, 57] and the density of trails inside a 500m buffer [54] around each camera site. We used a 500m buffer to have an overall understanding of the recreation impact around each camera trap site, representing a trade-off between the relatively large scale of movement by focal mammal species using a site. Previous research suggests that variation in buffer size (i.e., spatial resolution) has less impact on understanding mammal habitat use than does the size of the study area (i.e., spatial extent), which is addressed by our large sampling array [58]. Density and distance measure the relationship between recreational features in a landscape (i.e., the potential intensity) and a camera site but do not measure the actual intensity of recreation activity. For this, we used data extracted from two social media platforms that track physical exercise, AllTrails and Strava [55].

AllTrails (https://www.alltrails.com) is a popular website in North America (and elsewhere) and aims to record and share trail information for outdoor enthusiasts. Users can leave reviews about their experience on a trail. We used the number of reviews per trail as an index of its intensity of use (i.e., we assumed that the number of reviews posted about a trail would be proportional to the number of times it was being used [59]). Within the AllTrails database, we searched for the areas of Bighorn and Castle, which returned spatial zones with an associated list of trails. We verified that the spatial zones and trails overlapped the areas sampled by our camera trap grids and adjusted the search area as required. Through this search we identified 71 trails for Bighorn and 67 trails for Castle. In March 2022, we visited the webpage for each

trail and recorded the trail name and number of reviews. We also downloaded the GPX track for each and converted them into a shapefile to map the position of trails relative to camera traps. We merged overlapping trails and summed the number of reviews for the component trails. We created a 500m buffer around each camera trap and extracted the number of reviews for all trails found within the buffer.

Similar to All Trails, Strava (https://www.strava.com) is a smartphone application made for users to record and share their physical activities [60]. Strava can be used to track any type of physical activities, but core users remain runners, hikers, and cyclists, and only running and cycling activities could be extracted for analysis at the time of this study (i.e., only non-motorized activities can be extracted). We extracted Strava data using their Python Application Programming Interface (API), which is freely available to anyone with a Strava account. We adapted the Python code developed by Stelmach and Beddow (2016) [61] to find all publicly visible segments uploaded to Strava by sequentially searching within our study areas at increasingly broad scales. Each segment can be associated with a specific activity that can be analyzed separately or, as in our case, together. For both Bighorn and Castle, we first searched within individual 0.5 x 0.5 degree grid cells, then we doubled the grid cell size and repeated the search. We continue to double the grid cell size until a single cell encompassed the whole study area. All duplicate segments found during the search were removed. Within each area and for each segment, we extracted the date of the segment creation and the total number of times the segment had been recorded by any user (i.e., the total number of efforts). For each segment, we created a normalized 'use rate' (effort per year) by dividing the total number of efforts for a segment by the amount of time since the creation of the segment. For those familiar with the Strava Global Heat Map (https://support.strava.com/hc/en-us/articles/216918877-Strava-Metro-and-the-Global-Heatmap), we note that while our methodology pulls raw data used in the heatmap, it may generate different results for a variety of reasons (e.g., the heatmap is updated monthly, low activity segments do not show up on the heatmap, some of the data in the heatmap may not be publicly available, etc.). However, since the quantitative values of the heatmap colors are not available, and the colorization technique is poorly described, we chose this approach as a reproducible method to generate quantitative dataAll Strava data were extracted using Strava's REST API (v3) and the stravalib library (v0.10.4) programmed in Python (v3.10.1). We used the same 500m buffers around each camera trap (as described for AllTrails above) and extracted any Strava value found within each buffer. Finally, we averaged all the Strava values found inside each buffer.

**Factors influencing mammal responses to recreation.** We characterized factors which we hypothesized could influence mammal responses to recreation (influencing factors), specifically: the designation trail, percentage of forest, study area (Bighorn or Castle), percentage human footprint, the type of management and season (see Table 1 for details and predictions). Since we were interested in whether and how these factors modulate the effects of recreation, we included them as interactions in combination with each recreation measure in our model framework (see section 2.4).

To test if the type of outdoor activity influenced mammal response to recreation, we extracted information about the designation of each trail. We used the designated trails dataset: a collection of linear features representing the location of trails on crown land administered under the Public Lands and Parks Acts in Alberta (https://geodiscover.alberta.ca/). These linear features were associated with various information about each trail such as their time of opening and their designations. We classified the area inside a 500m buffer around each camera traps depending on the presence of trails and their use by recreationists. Therefore, each camera trap location was given a classification "motorized", "non-motorized", or "no-trail". However, some trails were closed to specific activities during some periods. We classified the

trail designation for each season (i.e., winter Dec 1—Apr 30, or summer May 1—Nov 30). If a trail was open to motorized activities for at least some part of a season, it was classified as "motorized" for that season. Similarly, if at least one out of multiple trails within the camera trap buffer were designated for motorized activity, we classified the site as "motorized" due to hypothesized higher level of disturbance from this activity relative to non-motorized.

The percentage of forest, as a proxy of the visibility, was obtained for each camera trap site (data source in S1 File). To avoid extracting value at a single geographic point, we created a 500m buffer around each camera and calculated the percentage of forest (coniferous, broadleaf and mixed forest were joined together) inside each buffer.

To calculate the percentage of human footprint for each camera trap, we used the ABMI Human Footprint Inventory (data source in S1 File). This represents anthropogenic disturbances on the Alberta land-base, obtained from SPOT6 satellite imagery. Inside each 500m buffer, we extracted and summed the length of the different anthropogenic features (e.g., roads, buildings, powerlines). We then calculated the percentage of area covered by these features inside each buffer.

The camera trap sampling spanned several different types of land use zones with varying recreation management regimes (Fig 1). To assess whether the general type of recreation management in a sampled area influenced the effects of recreation on mammals, we created an ordinal "management" variable that broadly categorized the level of restrictions on outdoor activities according to land use policies. The ranked values of this variable spanned a gradient from no special management i.e., public land with few restrictions (value 1), to areas managed for higher levels of recreation (Public Land Use Zones, Public Land Recreation, Provincial Recreation Areas; value 2), to areas with more protection of natural or "wilderness" features and thus more restrictions on recreation (Provincial Parks; value 3, Wildland Provincial Parks; value 4). Camera trapping was not stratified or evenly distributed across the land use zones. Public Land Use Zones (PLUZs) are delimited for the management of recreational land use, with each zone having rules that permit or prohibit certain activities depending on the time of the year or local conditions such as snow fall. Public Land Recreation (PLREC) zones allow high recreational use, and Provincial Recreation Areas (PRAs) are typically small areas surrounding campgrounds or day-use sites. Provincial Parks (PP) protect both natural and cultural landscapes and features and include a range of visitor facilities, while Wildland Provincial Parks (WPP) are more remote and free of any modern construction (no significant infrastructure). The western and southern portions of the Bighorn study area are mostly composed of six PLUZs, with some PLRECs and PRAs, while the eastern portion is composed of lands without any specific type of recreation management (i.e., public land; Fig 1). The Castle study area is composed of Castle Provincial Park (PP) and Castle Wildland Provincial Park (WPP), with some smaller portions of PLUZ (Fig 1).

**Additive landscape factors and collinearity.** We included other variables in our model that represented landscape factors known to influence species space use (and thus species detections), but which we did not expect to influence the effects of recreation (i.e., in our study we do not have predictions for theses variables to influence species' responses to recreation). These variables were land cover type, vegetation productivity, elevation, and distance to surface water (Table 1).

The land cover type (i.e., dominant vegetation type) was obtained using the same dataset as the percentage forest above. We extracted for each location a category of vegetation: Developed (e.g., roads, power line), Forest, Grassland-Shrubland, or Others. The land cover type and the percentage of forest, together, provide information about the habitat directly at the camera trap location as well as in its surrounding vicinity. We obtained information about vegetation productivity using the mean NDVI inside the 500m camera trap buffer. NDVI information

was obtained using the MODIS of the Terra satellite: MOD13A1 Version 6. This provided NDVI values at a per pixel basis at 500 meter spatial resolution [62]. For each camera trap, we calculated the elevation using the R package Elevatr [63]. Finally we obtained information about water stream and water body using the website https://www.altalis.com/, and we calculated the distance between the nearest water source and each camera using the function st_distance from the R package sf [64].

We tested for collinearity amongst our 13 predictor variables (Table 1). First, we tested the collinearity amongst the different quantitative variables using the Pearson correlation coefficient, using $|r| > 0.7$ as a threshold above which one of the correlated pair of variables would be removed (S1 Fig). However, as we used a mixture of continuous and categorical variables, we also used a Factor Analysis of Mixed Data (FAMD) to assess the similarity between all of our predictor variables [65] (S2 Fig). We subsequently removed the AllTrails variable from further analysis due to its close correlation with both the Strava variable and the density of trails, and its lower importance based on FAMD weightings in relation to those variables. We kept the Strava variable rather than the AllTrails variable as it has been previously used in multiple studies [32, 60, 66]. Since the management zones differed between the Bighorn and Castle study areas, our management type and study area variables were closely related, and we thus removed the categorical study area variable from further analysis (since it had only 2 categories vs. 4 categories used to delineate management differences).

## Modelling framework

We modelled patterns of space use by the focal mammal community in relation to the hypothesized predictor variables using the Hierarchical Modeling of Species Communities (HMSC) framework within the package *Hmsc* [67] in R v.4.0.2 [68]. The HMSC framework uses Bayesian inference to fit a multivariate hierarchical generalized mixed model [69]. This family of joint species distribution models relates coefficient estimates to hypothesized predictors through a regression framework, while quantifying species co-occurrences through random effects [67]. We modelled the site-by-season species count data using a lognormal Poisson distribution as suggested in Ovaskainen and Abrego (2020) [69].

We used interaction terms to test our hypotheses about factors that might influence species responses to recreation; i.e., we set up interactions in the HMSC model between each of the "influencing factors" and each of the "recreation variables" (Table 1). We included the other landscape variables as additive (non-interactive) effects in the model. We also used the number of days each camera trap was functioning as an additive effect to account for variation in sampling effort among camera traps (since not all camera traps were active over the entire periods of deployment, due to issues like battery failure or other malfunction). This is often accounted for using an offset or by creating a detection rate (the number of detections per season divided by the number of active days per season) as a response variable. However, the current version of the HMSC model implementation does not support offset configuration and only supports discrete variables. Camera trap station ID was included as a spatial random effect to account for non-independence between winter and summer counts at the same camera site.

Our HMSC model was fitted with four MCMC chains, each composed of 101000 iterations with a thinning interval of 100 and a burn-in length of 1000. Parameters were confirmed to have converged (i.e., chains mixed well) through visual inspection of trace plots, examination of effective sample size, and use of the Gelman and Rubin's Potential Scale Reduction Factor (PSRF) for which an approximate convergence is diagnosed when the upper limit is close to 1. We used pseudo-$R^2$ as a measure of model fit, the pseudo-$R^2$ is computed as squared Spearman correlation between observed and predicted values, times the sign of the correlation [69].

We used the *Hmsc* package to partition the variation explained by the fixed effects (i.e., predictor variables in Table 1). We evaluated the relative importance of groups of related predictors by summing variance for individual predictors [69], using the following groups: Recreation (Strava, trail density, distance to trail); Influencing factors (designation trail, percentage of human footprint, season, percentage forest, management type), Recreation Interaction (Recreation in interaction with Influencing factors), and Landscape (additive effects of elevation, water, land cover, NDVI; Table 1).

We calculated variation partitioning for each fixed effect and each species and scaled the variation explained by each variable (or group) by the total variation explained in the model (pseudo-R$^2$). We estimated the 95% credible intervals (CI) around the mean of the posterior distribution of each fixed effect for each species [69]. Due to the large number of results, we only focus our interpretation on effects with strong support, i.e., 95% CI not overlapping zero. We refer to conditional effects of "influencing factors" every time the interaction coefficient of one of these variables with a "recreation variables" was deemed well-supported. We investigated differences between responses of predator and prey species *post-hoc* by visually evaluating the responses of species within these guilds; although we note that future research can attempt to incorporate species traits into the HMSC modelling framework.

## Results

The total sampling effort was 90,467 camera trap days, with 52,188 days in Bighorn and 38,279 days in Castle. We obtained 16,833 independent detections of 34 mammal species (see S1 Table), from which 554 detections were removed due to either a) an inability to identify animal detections to the species level or b) a species having too few detections to achieve adequate model convergence (<60 independent detections; see S1 Table). This left 15 species with a sufficient number of detections to model (i.e., model effective sample size ≥ ~100 and PSRF ~ 1 (approximate convergence; see S1 Table).

### Relative importance of groups to explain mammals' space use

The HMSC model explained between 10–65% of the variation in patterns of space use across camera sites for the 15 species (Fig 2, top). Recreation on its own–as measured by the three different indices (distance, density, Strava; Table 1)–explained an average of only 9% of the variation in space use across species (range = 3–13%; Fig 2, bottom), and no species showed significant positive or negative responses to individual recreation variables (i.e., 95% CI overlapped 0; S2 Table). However, interactions between recreation and the hypothesized influencing factors (Table 1) explained on average 45% (19–61%) of mammal space use—considerably more than recreation main effects. Thus, in combination, the main and influencing factors of recreation explained considerable variation (Fig 2, bottom). The interacting factors on their own explained on average 14% (4–22%) of space use (Fig 2), with species more likely to be detected in areas with stricter management for some species in winter (S3 Fig—e.g., mule deer). The additive landscape variables explained an average of 6% (2–12%; Fig 2), including negative associations with elevation and grassland-shrubland habitat for several species (S3 Fig —e.g., gray wolf).

### Influencing factors of mammals' responses to recreation

We first looked at the interactions between the group of recreation measures (distance, density and Strava) and hypothesized influencing factors (Table 1). The interaction with management type explained the most variation in species space use: an average of 19% (10–30%; Fig 3A). The trail designation (motorized or non-motorized) also had an important interaction with

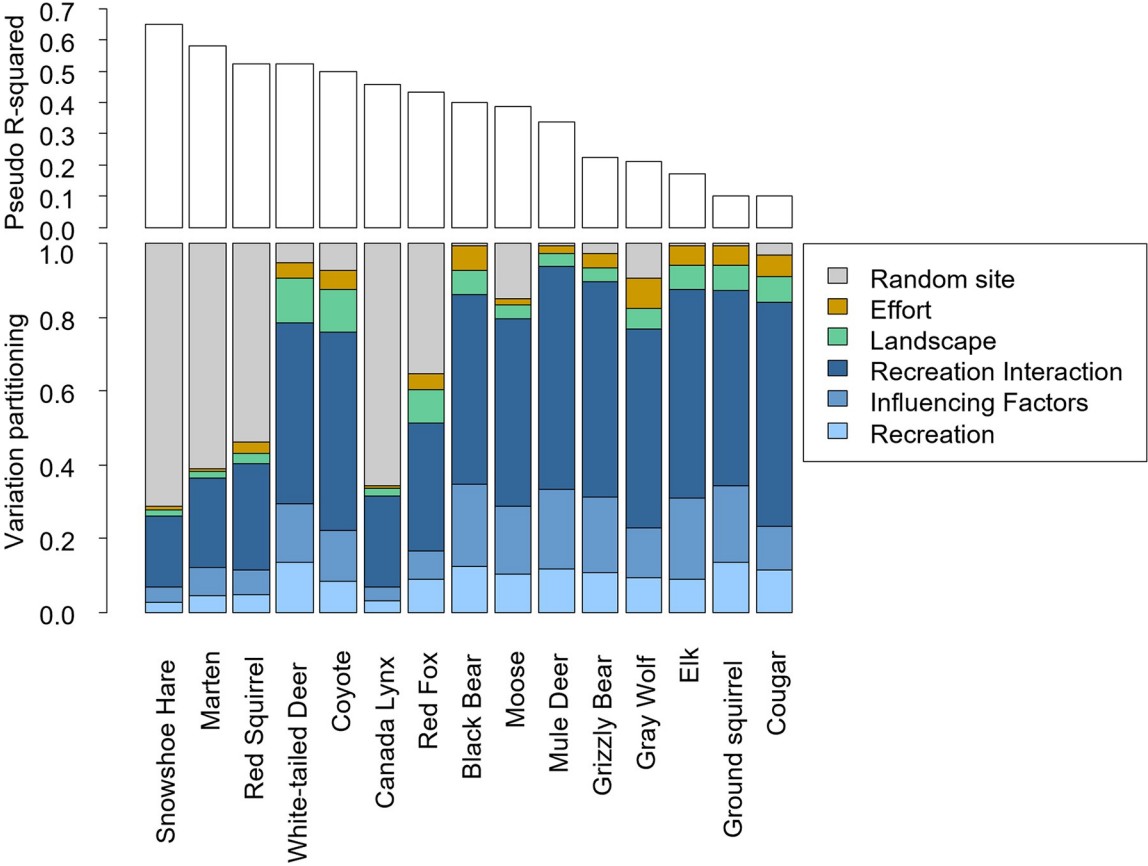

**Fig 2.** Variance explained (pseudo $R^2$, top graph) and proportion of explained variance (bottom graph) for each species. "Random" represents the variance accounted for by the station level random effect; "Effort" represents the variance accounted for by the camera deployment length; "Landscape" includes: Elevation, Distance to water, Land Cover and NDVI; "Influencing Factors" includes the trail designation, % forest, % human footprint, Management Type, and Season; "Recreation" includes the main effects of distance, density and Strava; "Recreation interaction" includes the interaction effects between each factor of "Recreation" and "Influencing Factors".

recreation activity in influencing mammal space use, accounting for on average 14% (5%-21%) of variation. The other variables (percentage forest, percentage human footprint, season) generally had weaker interactions with recreation, accounting for an average of less than 5% of the total variation across species (Fig 3A).

## Trail density

At the finer model resolution of individual recreation variables, trail density explained the most variation in species space use. Trail density explained on average 11% (3–24%) of variation across species (Fig 3B) when considering both main and interactive effects. The strongest interactions were between trail density and a) management type and b) trail designation (Figs 3–5). For most mammal species (excepting mule deer and ground squirrel) there was a negative interaction between trail density and management type, such that mammals were less likely to use sites with higher trail density in wilder areas with more restricted recreation (i.e., provincial and wilderness park lands; Fig 5A). For most species, there was a positive interaction between trail density and trail designation, although the specific responses to motorized and non-motorized trails varied among species, resulting in a lack of clear signal (Figs 4 and 5B). Some species increased their use of higher trail density areas when the trail designation

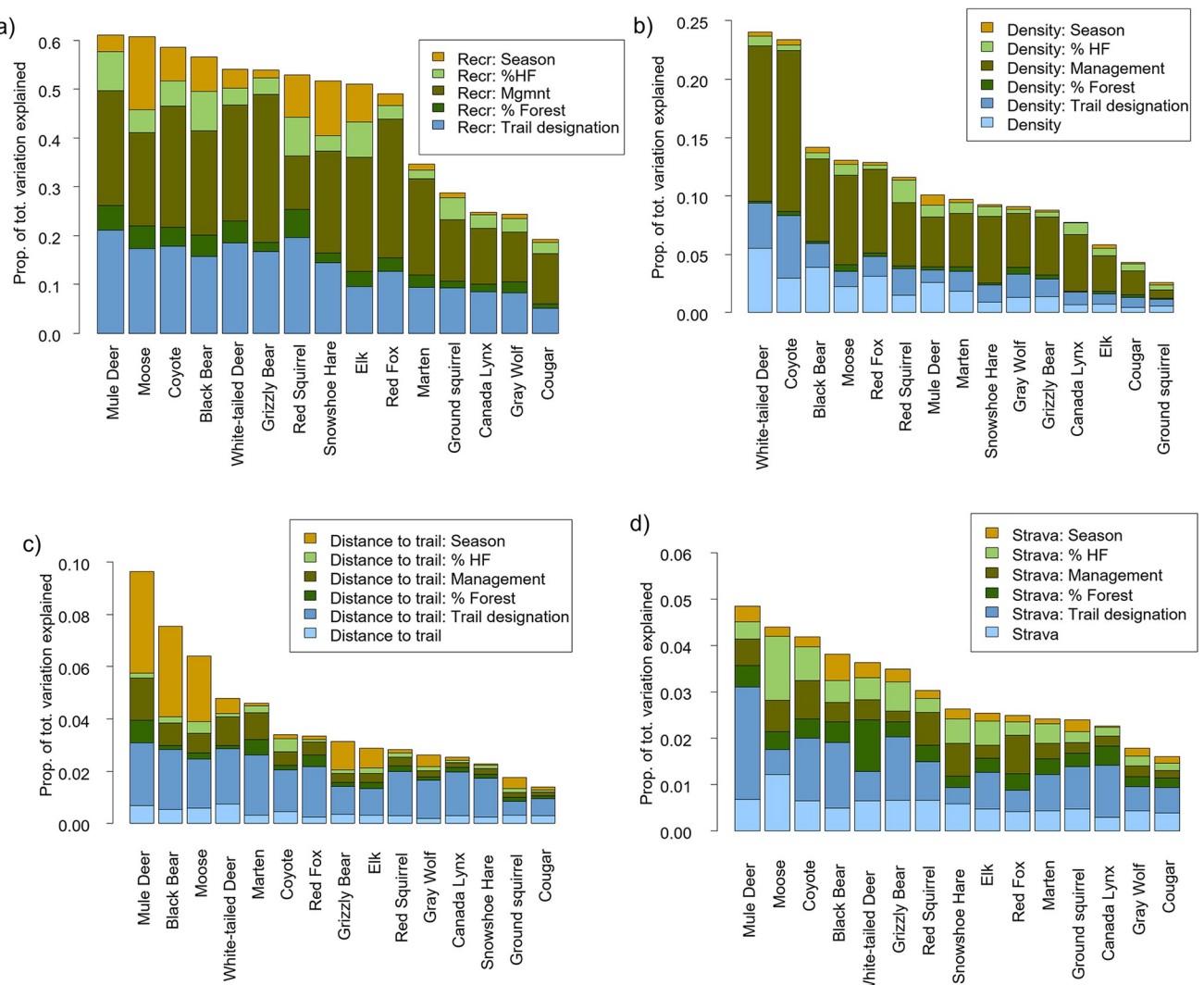

**Fig 3. (above) Proportion of total variation in species detections.** The proportion of total variation was obtained by multiplying the variation partitioning by the corresponding pseudo R2 for each species (see Fig 2). In a) we combined the proportion of total variation of Strava, distance, and density with interaction factors. In b), c) and d) we showed, respectively, the results of the density, distance, and Strava measures in interaction.

was motorized activity (e.g., wolf, cougar, snowshoe hare), some when the trail designation was with non-motorized activity (e.g., wolf, black bear, white-tailed deer), and some did not show this effect. For moose, use of areas with higher trail density was higher with increasing forest percentage, whereas for lynx and red squirrels, it was higher with increasing human footprint (Fig 4).

**Distance from trail.** The distance to the nearest trail explained on average 4% (1–9%) of variation in species detections, when considering both main and influencing factors (Fig 3C), with the strongest interactions with season. Space use for seven species showed a negative effect of interaction between distance and season (Fig 4): wolf, lynx, black bear, grizzly bear, moose, white-tailed deer and mule deer were found closer to trails during winter relative to summer (Fig 5C). Mule deer occurred further from trails where there was more forest, and closer to trails in areas with more restrictive management (e.g. park lands vs. public lands). Coyote occurred further from trails where there was more human footprint (Fig 4).

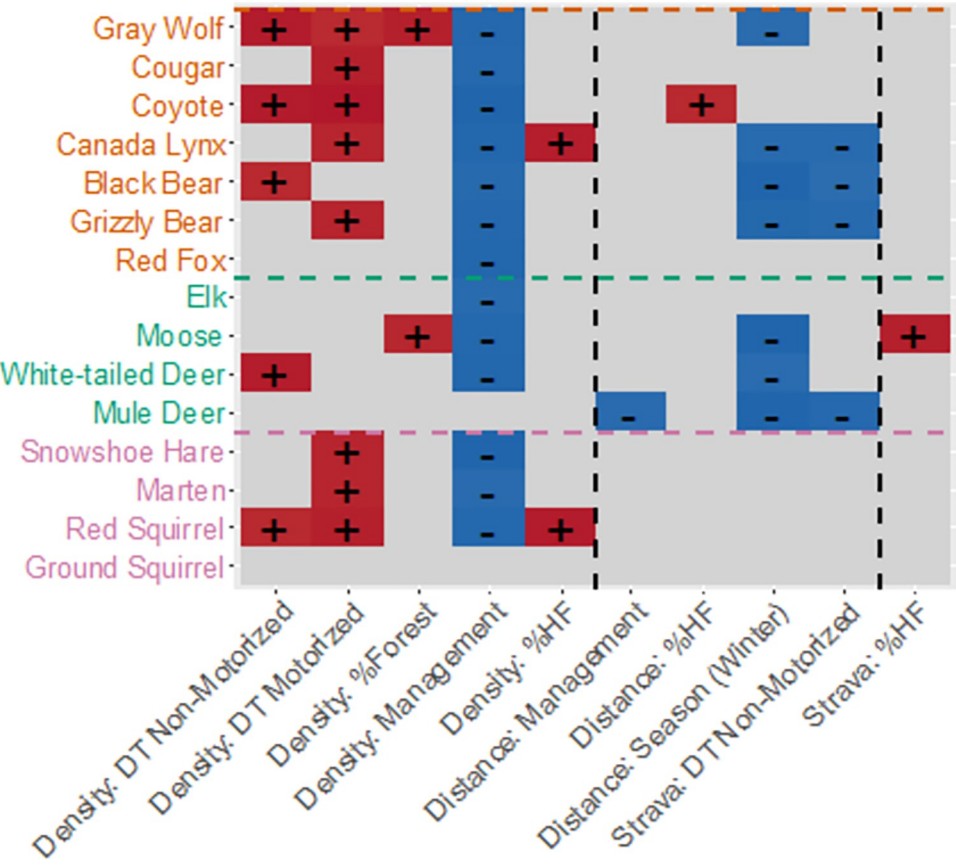

**Fig 4. Subset of species responses to recreation with interacting factors.** We only show responses for interactions that were significant for at least one species. Effect coefficients with at least 95% posterior probability above zero are shown in red with a "+", or below zero in blue with a "-". Predators are in orange, ungulates in green and small mammals in pink. Vertical lines separate the different recreation measures. DT = Designation Trail, HF = Human Footprint. For full estimates and confidence intervals, see S3 Fig.

**Recreation intensity (Strava).** The intensity or frequency of recreation activity around a camera site, as measured by our Strava variable, explained on average 3% (2–5%) of variation in species detections as a main effect and in interaction (Fig 3D). The interaction between Strava and the designation of trail was strongest. Specifically, lynx, black bear, grizzly bear, and mule deer, showed more negative responses to higher levels of Strava use in areas with trail designation for non-motorized activities (Figs 4 and 5D). Moose were more positively associated with Strava use in areas with higher human footprint. However, it is worth noting that Strava was only a measure of the intensity of non-motorized activities.

## Discussion

There have been numerous studies examining the effects of outdoor recreation on wildlife, with many different types of responses being documented [70–73]. Our study aimed to elucidate not only the responses of multiple mammal species to recreation in two partially protected regions in western Alberta, but also the degree to which responses were influenced by factors such as local management restrictions, trail designations, and environmental conditions. Using a hierarchical Bayesian community-level modelling approach, we found that mammals did not have consistently strong negative or positive associations with measures of recreation

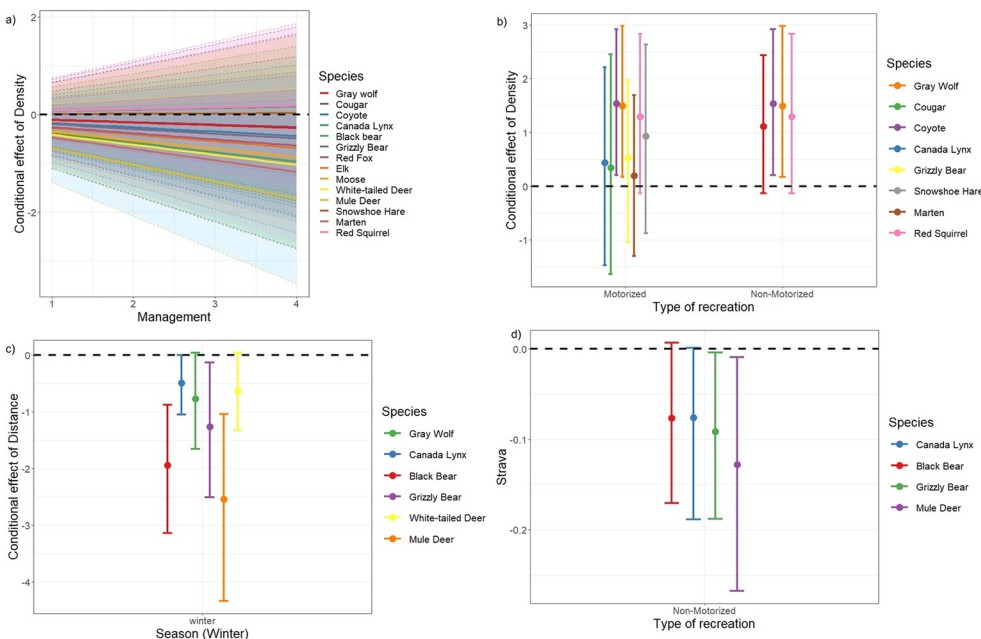

**Fig 5.** Conditional effects of a) density in interaction with the type of management. Values of management reflect the level of wilderness and recreation management in our study areas: public lands (value 1), Public Land Use Zones (PLUZs,) PLREC Public Land Recreation (PLREC), Provincial Recreation Areas (PRAs) (value 2), Provincial Park (PP) value 3 and Wildland Provincial Park (WPP) (value 4), b) density in interaction with the type of recreation, c) distance in interaction with the season, and d) Strava in interaction with non-motorized activities. Species represented with a response with at least 95% posterior probability (see Fig 3).

distribution and intensity, but that recreation interacted with other influencing factors to explain considerable variation in patterns of space use across multiple species. More specifically, we did not find evidence that predator and prey species consistently differed in their responses to recreation (hypothesis 1). We also found that the trail designation (motorized vs non-motorized) influenced species responses to recreation pressure, but that there were not consistent differences in responses across species (hypothesis 2). We also found that most mammals were less likely to use sites with higher trail density when they occurred in areas with more restrictive management of recreation activities (e.g., parks and wilderness parks; hypothesis 5).

Our two study areas—the Bighorn and Castle regions—are both experiencing increasing amounts of visits by recreationists (while there is a control over the number of trails), raising concerns about potential negative effects on wildlife. We did not directly compare recreation impact on mammals' space use in the two different areas, but instead used other landscape and influencing factors which are specific to these two areas (e.g., recreation management). It is encouraging that the camera trap surveys detected a diverse mammalian assemblage across these regions, and that we did not detect strong negative associations with recreation variables in regard to a possible avoidance of site with increased recreation pressure. It is also interesting that mammal responses to recreation appeared to be influenced by recreation management.

Our sampling covered a gradient of recreation management types across the two study areas, from public lands and recreation areas easily accessible to recreationists, to more remote park lands with a greater emphasis on protection of nature. We identified an interaction between the potential recreation activity at a sampling site—measured as trail density within a 500m circular buffer around a camera—and the recreation management, such that sites with a

higher density of trails were less used by mammals in the wilder park lands. In such areas, mammals are generally less likely to encounter recreationists in the broader landscape, and may thus be more sensitive to, and accordingly avoid, localized sites where more recreation occurs [54]. By contrast, mammals in areas where recreation activity is more ubiquitous may become habituated to humans and less likely to show this avoidance response [15]. Habituation and avoidance behavior can be altered by hunting [24, 74], a factor not accounted for in our study due to a lack of information on spatial variation in hunting pressure. However, previous studies have demonstrated its influence on species responses to recreationists [22, 24].

We only detected an interaction between recreation and management for trail density, and not for distance to trail or our Strava indicator of recreation activity, with the exception of mule deer which were found closer to trails in park lands (vs public lands). The physical presence of multiple trails in these remote areas might represent a visual cue that leads to avoidance (or attraction) by wildlife [75]. We attempted to incorporate visibility into our models by using the percentage of forest within a 500m buffer around a site, but this did not influence most species responses to recreation and may have been too coarse a measure. Moose and mule deer were exceptions, but with two different responses, with moose using areas with higher trail density more often when there was more forest cover (consistent with the notion that cover provides security), while mule deer showed higher use of areas further from trails where there was more cover (potentially consistent with a preference for longer sightlines to avoid human disturbance). We recommend further research into understanding how recreation management interacts with recreation activity, and in turn with forest cover, to affect wildlife. This has important implications for management policies, including whether and where to prioritize actions that limit recreation infrastructure (e.g., trail density) vs. those that limit recreation activity (e.g., number of hikers allowed in an area per time period).

Interestingly, we did not find a strong effect of percentage human footprint on most focal mammal species, either on its own or as an interaction with recreation (hypothesis 4). There were some exceptions where significant interactions between percentage human footprint and recreation were identified in our model. Lynx and red squirrels used areas with higher trail density more often when there was more human footprint, and similarly moose were positively associated with recreation activity measured by Strava in areas with more human footprint. Both patterns are consistent with habituation to recreation in more developed areas. However, in the case of red squirrels, the lack of avoidance might indicate an inability to avoid disturbance rather than habituation to stress and recreation due to their small home range. By contrast coyotes occurred at sites further from trails in areas with higher footprint, suggesting they may have been warier of people in more developed areas. Other studies have concluded that the amount of human footprint on a landscape is a major driver of species space use [76]. The mixed results in our study highlight the complexity and variation in different species responses to the same landscape contexts [33], but the somewhat limited responses to footprint may also reflect the fact that our sampled landscapes had relatively low levels of development overall. We recommend that future research extend our efforts by applying standardized camera trapping sampling across a larger number and gradient of landscape contexts [77].

As expected, the type of recreation activities permitted in an area around sampling sites (i.e. by trail designation) influenced mammal responses to recreation. We identified an interaction between recreation activity level (measured by Strava) and non-motorized trail designation for four species (lynx, black and grizzly bear, mule deer), such that they used areas with higher Strava activity less when nearby trails were designated non-motorized. However, this result may be due to the fact that our Strava data primarily represents non-motorized activity. On trails that allow motorized activity, Strava data may not reflect recreation intensity as well, which may explain why there was not a significant interaction between Strava data and

presence of trails designated as motorized. Thus, our finding may not actually indicate that non-motorized recreation has a stronger effect on these four species compared to motorized activity, but rather that we do not have an accurate representation of motorized recreation intensity in our dataset. Motorized activities are predicted to have a stronger impact on wildlife due to their higher levels of noise and speed [15], but recent reviews have reported that evidence of negative effects on wildlife from non-motorized activities is stronger than for motorized [70]. Motorized activities can have other impacts on the environment, such as soil erosion [78], and may cause impacts at larger spatial scales [70, 79]. They may also create wider travel corridors that are selected by some species (e.g., wolves, [80]). Thus, at local scales, non-motorized activities might affect behavioral responses but at landscape scales motorized activities are likely to have more impact on wildlife. In future research, it may be important to distinguish recreation activities at a finer resolution and obtain accurate estimates of motorized and non-motorized activity intensity, for example using Strava data it is possible to distinguish biking and running activities. Naidoo & Burton (2020) [12] found that mountain biking caused as much disturbance to wildlife as motorized activities, and more than other non-motorized activities like hiking and horseback riding. They also found that wildlife displacement could occur at finer temporal scales, and other studies have documented differences in wildlife responses to recreation during the day vs. at night [23, 81], such that areas with more recreation activity may be avoided during the day but used for movement under the cover of darkness [82, 83]. We recommend that future studies consider both spatial and temporal responses by wildlife and assess recreation activities based on the attributes that may underlie their impacts, such as speed, noise, and visibility.

Contrary to our prediction, we did not identify strong or consistent differences in responses to recreation between predator and prey species (hypothesis 1). The human shield hypothesis predicts that prey will be found close to human activity to protect themselves from predators which avoid human activities [26, 27]. We did not find evidence to support this hypothesis, as prey species were not consistently associated with our measures of recreation, nor did predators consistently avoid recreation. Predator-prey interactions are complex and human activities can lead to cascading effects on species interactions due to the impact on individual fitness and change in population demographics [29]. Similarly, predator-prey interactions are dynamic in time [84], which our study did not investigate as we only considered seasonal spatial patterns and not fine-scale temporal patterns. With more direct measurement of recreation activity—such as with camera traps placed on recreation trails, or using acoustic soundscape monitoring—finer-scale analytical methods such as avoidance-attraction ratios can be used to assess interactions between humans, predators, and prey [12, 73, 85]. Our results did suggest that smaller mammals were generally less responsive to recreation, despite indications from elsewhere that space use by small mammals may be affected by recreation, particularly due to snow compaction in alpine areas during winter [86]. Similarly, we did not conduct a fine-scale temporal analysis that would enable the identification of specific times when trails and areas with high recreation intensity were used by mammals. Mammals' space use in high recreation intensity areas could result from a shift in their diel activity cycle, avoiding recreation during the day and utilizing the trail network at night [23, 87].

We were unable to directly measure recreation in our study areas (since camera traps were not placed on recreation trails), but rather used multiple indirect measures of potential recreation pressure on wildlife habitats. Previous studies have often only used one measure of recreation, such as the presence or number of recreationists, or the times of their peak activities [88]. We found interesting differences in species associations with the multiple measures of recreation. For instance, more variation in mammal detections was explained by an interaction between trail density and management type than by an interaction between distance to trail

and management type. In contrast, an interaction between distance to trail and season explained more variation in space use for predators and ungulates, with these species being found closer to trails during winter (hypothesis 6). This result aligns with those of Gese *et al.* (2013) [6] and Bunnell *et al.* (2006) [89] that suggested recreation trails can facilitating animal movement in deep snow (e.g. trail compaction by skiing, snowmobiling [82]). It may also reflect more avoidance of trails by mammals in summer, when trails are busier, or more sensitivity due to reproduction statue (e.g., presence of young). Ultimately, the different patterns that we observed for different indirect measures of recreation could represent true variation in animal responses to different components of a "recreation landscape", or they could reflect noise in the relationships between the measures and the true underlying aspects of recreation to which animals are responding. Eventually, more research is needed to test and compare different indicators of recreation pressure, including cost-effective indirect measures like those used in our study, and more labour intensive but direct monitoring of recreation activity, such as camera traps. Validation of indirect methods (e.g., Strava use) with direct ground-based monitoring is desirable to better understand their accuracy and limitations in different contexts, and critical before relying on them as a primary monitoring tool. Encouragingly, the more direct monitoring methods are being aided by the incorporation of machine learning tools, for example to process of camera trap images [85]. However, validation will require more purpose-designed sampling. For example, in our study we could not directly validate the Strava data with camera traps since cameras were placed off recreational trails. Likewise, we recommend that future research integrates various measures of recreation into a unified metric to comprehensively grasp how the spatial distribution (e.g., density) and the intensity of recreation (e.g., Strava) collectively influence mammals' responses. Similarly, the proportion of trails used for a specific activity might affect the nature and strength of species responses to recreation.

There are additional opportunities to build on our study in future research. We used a community-level analytical approach to evaluating multispecies responses to recreation, in order to move beyond the inefficiencies of single-species case studies. However, the computational complexity of this modelling approach necessitated that we remove less common species from our analysis. These rare and elusive species, such as wolverine (*Gulo gulo*; [90, 91]), are usually of conservation interest but are difficult to assess in individual studies due to low detection rates. Continued monitoring over the long term will increase sample sizes for such species and confirm their continued presence—or conversely their extirpation from—recreation landscapes. Camera trap networks such as in National Parks and across ABMI sites can help meet this aim [92]. Similarly, these networks can aid in studying interactions between recreation and mammals at a large spatial scale. At an even larger spatial scale, comparing the impacts of recreation on mammals across continents that vary in the intensity and type of recreation activity (e.g. North America vs. western Europe) could potentially enhance our overall understanding of this subject. There are additional factors that may affect wildlife responses to recreation that we did not consider in this study. For example, species risk perception has been shown to depend on their long-term relationship with humans [93] and degree of habituation to human presence [25]. This phenomenon is difficult to quantify [44] but is a key factor in understanding the responses of species (and populations) to recreation. Other species traits such as relative brain size (as an index of cognitive ability [94]) and habitat or dietary breadth [7] could also influence species sensitivities to recreation. Such traits could be incorporated into modelling frameworks such as the one we used [69] in attempts to build on our work and explain more variation in species responses to variation in recreation activity.

## Conclusion

Recreation activity is increasing globally [1, 2], and there is a need to develop management strategies that support the benefits of recreation while limiting any negative impacts on wildlife populations [95, 96]. Our results suggest that mammal communities are not transversally impacted by recreation activities (i.e., there were not strong or consistent responses across species), and that observed impacts were not easily decomposed into base ecological differences such as between predator and prey guilds. More research is needed to understand whether and how species traits influence sensitivity to recreation. Such knowledge could help guide multi-species management approaches, or identify where species-specific approaches are needed.

When studying the impacts of recreation on species, we must not only look at the sources of recreation disturbance, but also consider how their effects may be influenced by different anthropogenic and landscape contexts, some of which have direct linkages to management decisions. We observed that space use by a community of mammal species in the Rocky Mountains of western Alberta was influenced by an interaction between recreation intensity and the type of recreation management, such that mammals showed more avoidance of sites in the vicinity of higher trail density within areas managed for wilderness values. We suggest that limiting the overall density of trails in these protected areas, or aggregating trails within a small portion of protected areas, may help to reduce conflict between recreationists and mammals. We also recommend more direct monitoring of recreation and its potential mechanisms directly affecting wildlife, such as frequency, noise and visibility. While behavioural responses by wildlife can provide early warning of impacts, we recommend linking these to monitoring of population-level responses, such as changes in survival or reproduction, to understand long term effects of human disturbances [36, 37]. Only by linking rigorous scientific monitoring to recreation management will we more fully understand the prospects for human-wildlife coexistence within the world's rapidly changing landscapes [97].

## Supporting information

**S1 Table. Summary of independent (30-minute threshold) camera trap detections (Dets) and proportion of camera trap sites with at least 1 detection (Prop) for each species in the Castle (n = 64) and Bighorn (n = 91) sampling areas.** Photo were obtained using the ABMI standard operating procedures protocol (Alberta Biodiversity Monitoring Institute. 2016). Also shown are the HMSC model diagnostics. Only species with a total of at least 60 independent detections (grey shaded) were included in the analysis, as there were problems with HMSC model convergence for species with fewer detections. The effective sampling size (Eff n) is the average effective sample size across all parameters for each species for each model chain. The Gelman and Rubin's Potential Scale Reduction Factor (PSRF) is measured across all parameters estimated for each species and assesses the distribution of samples from model chains; as a rule of thumb, the mean should be <1.1 to indicate model convergence. (DOCX)

**S2 Table. 95% Bayesian credible intervals from the HMSC model.** (DOCX)

**S1 File. Data Sources in relation to Table 1 in the main text.** (DOCX)

**S1 Fig. Pearson correlation coefficients between quantitative variables used in the HMSC model to assess the influence of recreation and other factors on mammal habitat use.** % HF: % Human footprint, Dist: Distance, Dens = Density. See Table 1 for more details about

variables.
(DOCX)

**S2 Fig. Factor Analysis of Mixed Data (FAMD) assessing the similarity between all our variables see Table 1 for more details about variables.** The left panel represents the correlation circle that shows the relationship between quantitative variables, the quality of the representation of variables, and the correlation between variables and the dimensions. The right panel represents the correlation between variables, quantitative and qualitative variables (in red), and the principal dimensions.
(DOCX)

**S3 Fig. Species responses to recreation, influencing factors, landscape variables and interaction between recreation and influencing factors with factors.** Responses with at least 95% posterior probability are shown in red with a "+" or in blue with a "-"depending if the relationship is positive (above 0) or negative (below 0). Predators are in orange, ungulates in green and small mammals in pink. Vertical lines separate the different recreation measures.
(DOCX)

## Acknowledgments

We thank Jason Fisher and Joanna Burgar for their important roles in the development of the research questions and methods. We thank members of the project steering committee for their support and feedback (Dallas Johnson, Dan Farr, Bonnie Drozdowski). Many staff at Innotech Alberta contributed to data collection and management (including Jason Fisher, Luke Nolan, Colin Twitchell, Greg Brooke). Members of the WildCo lab at UBC provided feedback on this research. Members of Jason Fisher's ACME lab at the University of Victoria also contributed to field work. We thank Don Livingston (Alberta Environment and Protected Areas), Dragomir Vujnovic (Alberta Public Lands), Courtney Hughes (Alberta Environment and Parks), Graham Wylde (Alberta Environment and Protected Areas), Wonnita Andrus (Nature Conservancy of Canada), Brad Tucker (Alberta Environment and Parks), and Chad Willms (Alberta Environment and Parks) for sharing knowledge about the Bighorn and Castle areas. Chris Beirne and Patrick Thompson provided advice on statistical modelling.

## Author Contributions

**Conceptualization:** Solène Marion, Gonçalo Curveira Santos, Emily Herdman, Anne Hubbs, A. Cole Burton.

**Data curation:** Solène Marion, Sean Patrick Kearney.

**Formal analysis:** Solène Marion.

**Funding acquisition:** Emily Herdman.

**Investigation:** Gonçalo Curveira Santos.

**Project administration:** Emily Herdman, A. Cole Burton.

**Supervision:** Emily Herdman, Anne Hubbs, A. Cole Burton.

**Validation:** Gonçalo Curveira Santos.

**Writing – original draft:** Solène Marion.

**Writing – review & editing:** Solène Marion, Gonçalo Curveira Santos, Emily Herdman, Anne Hubbs, Sean Patrick Kearney, A. Cole Burton.

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
