## [Decision Letter · Decision Letter 0]

21 Sep 2023

PONE-D-23-21203Mammal responses to human recreation depend on landscape contextPLOS ONE

Dear Dr. Marion,

Thank you for submitting your manuscript to PLOS ONE. After careful consideration, we feel that it has merit but does not fully meet PLOS ONE’s publication criteria as it currently stands. Therefore, we invite you to submit a revised version of the manuscript that addresses the points raised during the review process.

We look forward to receiving your revised manuscript.

Kind regards,

Bogdan Cristescu

Academic Editor

PLOS ONE

“The project was funded by Alberta Innovates, Innotech Alberta, and Alberta Environment & Parks (now Alberta Environment and Protected Areas). Additional support for this research was received from the University of British Columbia and the Natural Sciences and Engineering Research Council of Canada (Canada Research Chair and Discovery Grant RGPIN-2018–03958 to A.C.B).”

“We thank Jason Fisher and Joanna Burgar for their important roles in the development of the research questions and methods. We thank members of the project steering committee for their support and feedback (Dallas Johnson, Dan Farr, Bonnie Drozdowski). Many staff at Innotech Alberta contributed to data collection and management (including Jason Fisher, Luke Nolan, Colin Twitchell, Greg Brooke). Members of the WildCo lab at UBC provided feedback on this research. Members of Jason Fisher’s ACME lab at the University of Victoria also contributed to field work. We thank Don Livingston (Alberta Environment and Protected Areas), Dragomir Vujnovic (Alberta Public Lands), Courtney Hughes (Alberta Environment and Parks), Graham Wylde (Alberta Environment and Protected Areas), Wonnita Andrus (Nature Conservancy of Canada), Brad Tucker (Alberta Environment and Parks), and Chad Willms (Alberta Environment and Parks) for sharing knowledge about the Bighorn and Castle areas. Chris Beirne and Patrick Thompson provided advice on statistical modelling.

The Wildlife CAMERA project was funded by Alberta Innovates, Innotech Alberta, and Alberta Environment & Parks (now Alberta Environment and Protected Areas). Additional support for this research was received from the University of British Columbia and the Natural Sciences and Engineering Research Council of Canada (Canada Research Chair and Discovery Grant RGPIN-2018–03958 to A.C.B).”

“The project was funded by Alberta Innovates, Innotech Alberta, and Alberta Environment & Parks (now Alberta Environment and Protected Areas). Additional support for this research was received from the University of British Columbia and the Natural Sciences and Engineering Research Council of Canada (Canada Research Chair and Discovery Grant RGPIN-2018–03958 to A.C.B).”

7. We note that Figure 1 in your submission contain [map/satellite] images which may be copyrighted. All PLOS content is published under the Creative Commons Attribution License (CC BY 4.0), which means that the manuscript, images, and Supporting Information files will be freely available online, and any third party is permitted to access, download, copy, distribute, and use these materials in any way, even commercially, with proper attribution. For these reasons, we cannot publish previously copyrighted maps or satellite images created using proprietary data, such as Google software (Google Maps, Street View, and Earth). For more information, see our copyright guidelines: http://journals.plos.org/plosone/s/licenses-and-copyright.

Additional Editor Comments:

Two expert reviewers have now provided their feedback and I invite you to answer their questions and suggestions point by point. Please clarify the field sampling strategy, as the infrequent camera trap check routine would have presumably resulted in many failures. The use of lure seems strange for a study that analyzes community responses of a variety of species including ungulates and smaller non-carnivorous mammals, which might presumably be deterred by the lure due to predation risk if the lure attracts carnivores. Also, there would be a decline in lure attractiveness which would be particularly pronounced given the infrequent camera trap servicing. Seasonal comparisons in the responses of mammals community could be affected by the season when the lure was placed, and when within the timeline of the season this occurred; timing of lure and addition of the lure might confound some of the effects of recreation on animal detection.

Regarding the modelling approach, pseudo R2 is not a well established metric of model fit, please consider alternatives. Please define conditional effects (presented in Fig. 5) in the Methods section.

Supplementary S5 Table 2 reveals a highly complex model output and suggests a possibly overparameterized model with a large number of parameter estimates recorded, and many interaction terms. Did you not run into model converge issues with this approach? I wonder how the all-inclusive approach you chose would compare to running the analysis in 3 sets of models - for each functional group (carnivores, ungulates, and smaller mammals)

It appears that you did not account for imperfect detection probability and instead opted to model detection rate (frequency of images). Further explanations would be needed as to why the approach you chose is superior to a multi-species occupancy modelling framework that builds a detection history to model imperfect detection.

Looking forward to a revised version.

Reviewers' comments:

Reviewer's Responses to Questions

**Comments to the Author**

1. Is the manuscript technically sound, and do the data support the conclusions?

Reviewer #1: Yes

Reviewer #2: Yes

2. Has the statistical analysis been performed appropriately and rigorously? 

Reviewer #1: Yes

Reviewer #2: Yes

3. Have the authors made all data underlying the findings in their manuscript fully available?

Reviewer #1: Yes

Reviewer #2: Yes

4. Is the manuscript presented in an intelligible fashion and written in standard English?

Reviewer #1: Yes

Reviewer #2: Yes

5. Review Comments to the Author

Reviewer #1: Here the authors use extensive camera trap datasets from two study areas in Alberta to examine the impacts of recreation on a range of mammal species. This is valuable work tackling a complex issue and is particularly notable given the authors’ attempts to capture multiple aspects of recreation and their interactions with other anthropogenic factors (e.g., land management). The paper is also quite well written and a compelling read. I have a few suggestions below that I hope will help bolster an already strong piece of work.

The authors make extensive use of a 500 m buffer area around each camera trap to summarize variables related to human use, forest cover, etc. They note that this buffer size was chosen to “have an overall understanding of the recreation impact”, but I’ll suggest that it would be helpful to provide a bit more justification for why this buffer size was deemed more appropriate than, say, 250 m or 1 km in providing such an understanding. As wildlife ecologists, I think we’re often tempted to just chose a single arbitrary buffer size when summarizing environmental covariates, but there is ample evidence from the landscape ecology world that this is actually an incredibly important assumption when quantifying species-environment relationships and that the buffer size chosen for a particular covariate can influence whether an effect is detected at all and even the directionality of that effect (i.e., positive vs negative influence of a covariate on species habitat use). See Jackson and Fahrig 2015 (Global Ecol Biogeogr, 24:52–63) for a very helpful (and troubling) review of this topic. All that said, I’m not suggesting there’s a need to go back and revisit the scale(s) at which your covariates were summarized. But it might be valuable to provide some additional justification for why the scale you chose is an appropriate one to apply across species and covariates.

Relatedly, for your land cover covariates, were these just extracted at the single pixel associated with a camera trap point location? Why not also summarize these variables at a broader scale (e.g., as % cover)? This would then correspond with the assumption you’re making with the 500m buffer, i.e., that animals are integrating information on land cover across a broader scale than just their immediate vicinity when deciding whether to use the habitat associated with a given camera trap.

Regarding the site-by-season species counts that were used as the response variable and summed across the full sampling period, I see that the authors included the number of days a camera was active as a term in the model, but, if there were substantial differences between the number of days cameras were active, I wonder if it might make more sense to turn your counts into a detection rate (i.e., number of detections per season divided by number of active days per season) to better account for differences in sampling period (which, if large, would add substantial amounts of noise to counts).

L288-295: I think I understand the motivation for classifying a given camera site as being associated with motorized trails if even a single trail w/in 500m was motorized. But I wonder if this complicates your related trail density estimate in any way. Did the authors consider calculating separate trail density estimates for each trail type (motorized and non-motorized), given the expectation of differential impacts between the two types? This would then seem to obviate the need for a separate categorical trail type covariate.

The authors do ultimately nod to the effects of diel cycle in driving wildlife responses to recreation in the discussion, but I wonder if they considered explicitly incorporating this in the analysis, e.g., by calculating counts (or rates…) by season and by day vs night. This could help explain several of this study’s findings regarding positive responses to high trail density. Other work (e.g., Nickel et al. 2020. Biol Cons, 241:108383) has shown that increased use of areas with high recreation intensity by many mammal species is also associated with increased nocturnality, presumably meaning wildlife are avoiding humans during the day but then using their trail networks at night. This could imply somewhat cryptic impacts on wildlife if temporal niches are constrained/forced to overlap.

L380: R^2 estimates variance explained but isn’t really a measure of model fit. You could have a poorly fitting model (e.g., the specified data generating distribution does not actually match the distribution of observed data) that still explains a reasonable amount of variation. Posterior predictive checks are typically the preferred way of examining fit for Bayesian models.

MINOR POINTS

L104-106: This statement could use a reference.

L132-145: The authors might consider restructuring their set of hypotheses as a numbered list. This could make it easier for the reader (and the authors) to refer back when interpreting the results (e.g., “we found support for Hypothesis 1…”)

L179-180: “A systematic random design was used in both areas…” Not clear what the random aspect of the design is. Is this not just systematic sampling?

L362: What are “multivariate response terms”? Do you mean coefficient estimates?

I hope you find this useful.

Best,

Justin Suraci

Reviewer #2: Review for PLOS ONE Mammal responses to human recreation depend to landscape context

I really enjoyed reviewing this manuscript, that I found very interesting, very well written and using an original modelling approach. Results are relevant and will certainly contribute to the growing body of evidence regarding animal responses to outdoor activities and recreation, a potential impact of growing presence and intensity across the globe, with increasing conservation concern. However there is one main point that the authors did not consider that I think might hamper the results presented in the manuscript. I also have several minor comments.

My main doubt regards the methodology used to extract the variable describing recreation intensity, namely the Strava variable. Corradini et al. 2021, that you cite, used a different approach to extract an index based on Strava data: they extracted and rasterised the Strava map, rather than using the number of reviews by the users. I am not sure that these two methods are equivalent, does the number of reviews by users really reflect the intensity shown by Strava in their maps? Furthermore, I have an even more serious concern regarding this point: how can you be sure that your index based on Strava data realistically reflects the intensity of recreation? This data certainly needs a validation! Corradini et al. 2021 validated their index using camera-trapping based on trails and found a positive correlation. But this positive correlation needs to be proven locally in every study area one may choose, since it is not guaranteed that Strava data will thoroughly reflect real on-the-ground frequentation of trails everywhere. To sum up: 1- I am not sure your method is equivalent to Corradini et al. 2021 one and equally representative; 2 – In any case this index should be validated against another method on-the-ground to make sure it gives a reasonably faithful picture of recreation in your area.

Other comments:

Lines 82-87: The whole introduction is interesting, well written and well documented. However here I think you lack at least to mention that also hunting is a major factor that can influence avoidance behaviours in mammals. Much research shows that fear of humans can be influenced by the presence and intensity of hunting in a specific area.

Lines 104 – 112: Here I would strongly suggest to discuss the potential long-term effects that recreation might have on wildlife species, communities and ecosystem functions. Even when we observe clear avoidance behaviors in mammals towards human recreation we do not know if this avoidance leads to decreased fitness and hence diminishing population trends. One study that evaluated this aspect is Salvatori, M., Oberosler, V., Rinaldi, M., Franceschini, A., Truschi, S., Pedrini, P., & Rovero, F. (2023). Crowded mountains: Long-term effects of human outdoor recreation on a community of wild mammals monitored with systematic camera trapping. Ambio, 52(6), 1085-1097. https://link.springer.com/article/10.1007/s13280-022-01825-w, whose results are relevant also for the interpretation of you results in this manuscript.

Additionally, I think that here or in the discussion another point that should be explored, also in relation to your results, is whether recreation can alter herbivory and predation patterns, resulting in direct consequences on other ecosystem components: a relevant subject also for protected area managers. One relevant work here is Di Nicola, W., Mols, B., & Smit, C. (2023). Human recreation shapes the local scale impact of ungulates on the carbon pools of a temperate coniferous forest. Global Ecology and Conservation, 46, e02574. https://www.sciencedirect.com/science/article/pii/S2351989423002093

Lines 183: Here you say that minimum distance from a trail was 50 meters but in Table 1 the minimum distances you report for the 2 areas are well below 50 meters.

Line 188: I do not understand why you lured your camera trapping sites. The lure effect will have faded a lot before your sampling was ended, since you kept your cameras for more than one year. Luring the site would in this case bias sampling with a higher detection probability at the start of sampling that would have decreased as time passed by. Luring could also introduce biases towards certain species more attracted to the scent, and since you are using community modelling I do not think this is a wise choice.

192: So you controlled cameras only once per year? Didn’t this result in many battery failure and SD card filling?

242: I wonder if one could merge the two types of info, distance from trail and recreation intensity on that trail into a single index weighting both factors together, since animals could remain more distant specifically to those trails more intensely visited by humans. If I am not wrong you did not try the interaction Distance:Strava.

265: This sentence is not clear: what do you mean by entire camera trap sampling area? The area sampled by one single camera trap or the whole study area?

Line 274: See my main concern regarding validation of the Strava data.

Line 370: Camera days can also be conveniently shaped as an offset.

Line 414: you accidentally put 2 ‘with’ at the end of this line.

Line 437: I would put ‘(see Figure 2)’ to avoid confusion.

Line 488: This last sentence is quite cryptic: why did you used it in combination with non-motorised trails then if it does not reflect vehicle passages at all?

Line 523: Again, I feel you here overlook the possible effect of hunting in determining avoidance behaviours.

Line 532-534: These two possible explanations seem completely in contrast one another.

Line 539: I wonder whether this lack of effect derives from the weakness of the Strava variable you extracted in representing real recreation. Alternatively, it could be that your study areas have relatively low recreation intensity rates, compared to more human dominated landscapes like those of western Europe. Comparing and discussing the responses to recreation between North America and western Europe could also be very interesting.

Line 546: What do you mean with this? What does this difference in home range size entail?

Lines 555 – 557: I did not understand what you mean here with leveraging, please clarify.

Line 622: ‘tpressure’ typo.

Line 658: ‘ewithin’ typo.

Line 664: Here you wisely mention the potential long-term effect on animal population demography potentially driven by avoidance of human recreation: see the paper I cited above that tackled this topic.

6. PLOS authors have the option to publish the peer review history of their article (what does this mean?). If published, this will include your full peer review and any attached files.

Reviewer #1: **Yes: **Justin Suraci

Reviewer #2: No

---

## [Author Response · Author response to Decision Letter 0]

11 Nov 2023

Additional Editor Comments:

Two expert reviewers have now provided their feedback and I invite you to answer their questions and suggestions point by point. 

*** Thank you very much for considering our manuscript, and for reviewing it yourself.

Please clarify the field sampling strategy, as the infrequent camera trap check routine would have presumably resulted in many failures.

*** Thank you for your comment. We clarified this in the method: “Cameras were visited once per year to retrieve photos and refresh batteries and lure. More frequent visits were not possible due to access limitations in the remote locations. We considered camera trap failures by examining the first and last days of daily timelapse photos, noting any days in which the camera was not functioning. We calculated the number of active sampling days for each camera trap and season to account for variation in sampling effort (see below). As a result, one camera trap with high failure rates was excluded from our analysis”. Line 206-212

The use of lure seems strange for a study that analyzes community responses of a variety of species including ungulates and smaller non-carnivorous mammals, which might presumably be deterred by the lure due to predation risk if the lure attracts carnivores. Also, there would be a decline in lure attractiveness which would be particularly pronounced given the infrequent camera trap servicing. Seasonal comparisons in the responses of mammals community could be affected by the season when the lure was placed, and when within the timeline of the season this occurred; timing of lure and addition of the lure might confound some of the effects of recreation on animal detection.

*** The camera trap arrays used in this paper are part of a larger monitoring program with multiple objectives, some of which aim to monitor and estimate abundance of elusive species such as rare mustelids (e.g. wolverine, fisher). Therefore, lures were used to maximize the chances of capturing those species. However, it is worth noting that species particularly sensitive to lure, such as fishers (Holinda et al. 2020), were not included in our analysis due to a lack of sufficient detections. Holinda et al. (2020) showed that responses to lures are highly variable across species, but they found that mammals were detected in almost equal numbers at camera traps (CTs) with and without a lure. In the same study, they also highlighted that seasonal patterns in detections are not masked or confounded by the deployment and attenuation of lure (that study used the same scent lure used in our study). 

Importantly, as the lures were applied at every camera trap, they didn't compromise our inference on the influence of factors on species spatial distribution. Any positive bias in species counts induced by the use of lure is common to all stations and we were interested in relative differences across space. We recognize that the response to lures might be species specific, but again here, we were not interested in comparing absolute counts across species but rather looking at species-specific use of space patterns.

We added the text: “Lure was used as part of broader monitoring objectives associated with this camera array, including detection of elusive species such as wolverines and fishers, which are particularly sensitive to lures [49]. However, due to few detections of these species, we did not include them in our analysis (see S1), and previous research indicates that this scent lure does not significantly impact detections of other species [49]. Moreover, as lures were applied consistently at every camera trap, they didn't compromise inference about the factors that we hypothesized to influence spatial heterogeneity in species detections. L198-205

[49]:Holinda, D., Burgar, J.M., Burton, A.C., 2020. Effects of scent lure on camera trap detections vary across mammalian predator and prey species. PLoS ONE 15, e0229055. 

Regarding the modelling approach, pseudo R2 is not a well established metric of model fit, please consider alternatives. Please define conditional effects (presented in Fig. 5) in the Methods section.

*** This metric is based on Ovaskainen O, Abrego N. Joint Species Distribution Modelling: With Applications in R. Cambridge: Cambridge University Press; 2020. doi:10.1017/9781108591720. We added more detail about this in the method: “the pseudo-R2 is computed as squared Spearman correlation between observed and predicted values, times the sign of the correlation”. L423-424

We added in the Methods about conditional effects: “We refer to conditional effects of “influencing factors” every time the interaction coefficient of one of these variables with a “recreation variables” was deemed well-supported” L437-438

Supplementary S5 Table 2 reveals a highly complex model output and suggests a possibly overparameterized model with a large number of parameter estimates recorded, and many interaction terms. Did you not run into model converge issues with this approach? I wonder how the all-inclusive approach you chose would compare to running the analysis in 3 sets of models - for each functional group (carnivores, ungulates, and smaller mammals)

*** Thank you for your comment: As described in the manuscript “Parameters were confirmed to have converged (i.e., chains mixed well) through visual inspection of trace plots, examination of effective sample size, and use of the Gelman and Rubin's Potential Scale Reduction Factor (PSRF) for which an approximate convergence is diagnosed when the upper limit is close to 1.” L419-422

It appears that you did not account for imperfect detection probability and instead opted to model detection rate (frequency of images). Further explanations would be needed as to why the approach you chose is superior to a multi-species occupancy modelling framework that builds a detection history to model imperfect detection.

*** Thank you for this relevant comment. The choice of using an HMSC approach over a multispecies occupancy modelling framework was not based on the idea of one approach being superior to another, but rather on the ecological signal of interest. While an occupancy framework models the presence-absence, or use – no-use, of camera sites during our sampling period, we anticipated that species responses to recreation could often be more subtle and related to changes in local abundance of animals and their movements or use of habitats features within their home range (Johnson’s (1980) third-order selection). 

While count data might be influenced by imperfect detection (Sollmann et al. 2013), it provides valuable ecological insights into how animals use the landscape relative to their surroundings (Stewart et al. 2018). It is important to note that detection rates are significantly affected by species-specific traits such as home range size, mobility, and body size. Therefore, our focus was on exploring spatial heterogeneity in species-specific counts rather than direct comparisons across different species. 

In addition, we also chose not to employ an occupancy model framework, which is often suggested to mitigate imperfect detection issues, because such models come with certain assumptions (such as site closure and independence of sites) that we found to be violated in our study. 

We added more detail in the text to reflect this rationale: “Count data provides crucial ecological insights into how animals navigate the landscape relative to their surroundings (Stewart et al., 2018). We chose not to adopt an occupancy model framework, commonly suggested to address imperfect detection (e.g., Sirén et al., 2021), as we believed that model assumptions would be violated (e.g., site closure, independence of sites), and, importantly, contend that the variation in detections reflects the ecological signal of interest - intensity of site use - rather than mere observation error (Neilson et al. 2018).” L214-220

Looking forward to a revised version.

Reviewer 1:

 Here the authors use extensive camera trap datasets from two study areas in Alberta to examine the impacts of recreation on a range of mammal species. This is valuable work tackling a complex issue and is particularly notable given the authors’ attempts to capture multiple aspects of recreation and their interactions with other anthropogenic factors (e.g., land management). The paper is also quite well written and a compelling read. I have a few suggestions below that I hope will help bolster an already strong piece of work.

*** Thank you very much for reviewing our paper, we appreciate your time and your feedback, we are glad you enjoyed reading our paper too. 

The authors make extensive use of a 500 m buffer area around each camera trap to summarize variables related to human use, forest cover, etc. They note that this buffer size was chosen to “have an overall understanding of the recreation impact”, but I’ll suggest that it would be helpful to provide a bit more justification for why this buffer size was deemed more appropriate than, say, 250 m or 1 km in providing such an understanding. As wildlife ecologists, I think we’re often tempted to just chose a single arbitrary buffer size when summarizing environmental covariates, but there is ample evidence from the landscape ecology world that this is actually an incredibly important assumption when quantifying species-environment relationships and that the buffer size chosen for a particular covariate can influence whether an effect is detected at all and even the directionality of that effect (i.e., positive vs negative influence of a covariate on species habitat use). See Jackson and Fahrig 2015 (Global Ecol Biogeogr, 24:52–63) for a very helpful (and troubling) review of this topic. All that said, I’m not suggesting there’s a need to go back and revisit the scale(s) at which your covariates were summarized. But it might be valuable to provide some additional justification for why the scale you chose is an appropriate one to apply across species and covariates.

*** Thank you for your comment and for including references. We appreciate your insight and recognize the significance of buffer size in ecological studies. Toews et al. (2017) emphasize that for studies of mammal habitat use, the spatial extent of sampling is more critical than buffer size per se. Our study prioritized a large spatial extent by encompassing two large landscapes, rather than pursuing an extensive evaluation of buffer sizes. To clarify this, we added the following statement: " We used a 500m buffer to have an overall understanding of the recreation impact around each camera trap site, representing a trade-off between the relatively large scale of movement by focal mammal species using a sites. Previous research suggests that variation in buffer size (i.e., spatial resolution) has less impact on understanding mammal habitat use than does the size of the study area (i.e., spatial extent), which is addressed by our large sampling array [Toews et al 2017]. " L269-274

Moreover, a full scale-sensitivity analysis was beyond the scope of this already-complex paper, and so we used a most parsimonious approach.

Toews, M., Juanes, F., Burton, A.C., 2017. Mammal responses to human footprint vary with spatial extent but not with spatial grain. Ecosphere 8, e01735.

Relatedly, for your land cover covariates, were these just extracted at the single pixel associated with a camera trap point location? Why not also summarize these variables at a broader scale (e.g., as % cover)? This would then correspond with the assumption you’re making with the 500m buffer, i.e., that animals are integrating information on land cover across a broader scale than just their immediate vicinity when deciding whether to use the habitat associated with a given camera trap.

*** Thank you for your comment. By incorporating the land cover type, we introduced a supplementary variable to the "percentage of forest" variable, which was calculated within the 500m buffer. We acknowledge that this might have been confusing as previously worded, and we added the following clarification: "The land cover type and the percentage of forest, together, provide information about the habitat directly at the camera trap location (small spatial scale) as well as in its surrounding vicinity (larger spatial scale)." L374-376

Regarding the site-by-season species counts that were used as the response variable and summed across the full sampling period, I see that the authors included the number of days a camera was active as a term in the model, but, if there were substantial differences between the number of days cameras were active, I wonder if it might make more sense to turn your counts into a detection rate (i.e., number of detections per season divided by number of active days per season) to better account for differences in sampling period (which, if large, would add substantial amounts of noise to counts).

*** As reviewer 2 mentioned, un-evven sampling effort across stations is often accounted by transforming the response variable into a detection rates or by using an offset; however, the current version of the HMSC model implementation does not support continuous response variables nor an offset configuration. We added “This is often accounted for using an offset or by creating a detection rate (the number of detections per season divided by the number of active days per season) as a response variable. However, offsets and continuous (rather than discrete) response variables are not yet supported in the current version of the HMSC package.” L412-416

L288-295: I think I understand the motivation for classifying a given camera site as being associated with motorized trails if even a single trail w/in 500m was motorized. But I wonder if this complicates your related trail density estimate in any way. Did the authors consider calculating separate trail density estimates for each trail type (motorized and non-motorized), given the expectation of differential impacts between the two types? This would then seem to obviate the need for a separate categorical trail type covariate.

*** Thank you for your comment. We wished to be able to distinguish the effects of these variables as we believe them to have different management significance, for example in terms of management actions to regulate the type of use on a trail vs. the density of trails in an area. By considering the interaction between trail density and trail type rather then having type-specific trail densities, we were able to explicitly test for such differences while still considering the possibility that species respond to the effect of the trail network regardless of recreation type (i.e. non-supported interaction). However, we agree with the reviewer in that the proportion of trails used for a specific activity might affect the nature and strength of species responses to recreation, which we discuss in lines 686-688: “Similarly, the proportion of trails used for a specific activity might affect the nature and strength of species responses to recreation”.

The authors do ultimately nod to the effects of diel cycle in driving wildlife responses to recreation in the discussion, but I wonder if they considered explicitly incorporating this in the analysis, e.g., by calculating counts (or rates…) by season and by day vs night. This could help explain several of this study’s findings regarding positive responses to high trail density. Other work (e.g., Nickel et al. 2020. Biol Cons, 241:108383) has shown that increased use of areas with high recreation intensity by many mammal species is also associated with increased nocturnality, presumably meaning wildlife are avoiding humans during the day but then using their trail networks at night. This could imply somewhat cryptic impacts on wildlife if temporal niches are constrained/forced to overlap.

*** Thank you for your comment; we agree that the shift in diel activities due to recreational activities could shed further light into the intricate responses of wildlife to recreation. Our study did not specifically aim to explore this aspect, as the focus of our paper on interactive impacts of recreation was already complex, and we thought adding yet another layer of intricacy would be confusing. However, we acknowledge the importance of discussing this point, and consequently, we have added a section addressing it in the discussion. “Similarly, we did not conduct a fine-scale temporal analysis that would enable the identification of specific times when trails and areas with high recreation intensity were used by mammals. Mammals' space use in high recreation intensity areas could result from a shift in their diel activity cycle, avoiding recreation during the day and utilizing the trail network at night”. L652-656

L380: R^2 estimates variance explained but isn’t really a measure of model fit. You could have a poorly fitting model (e.g., the specified data generating distribution does not actually match the distribution of observed data) that still explains a reasonable amount of variation. Posterior predictive checks are typically the preferred way of examining fit for Bayesian models.

*** See response to editor above

MINOR POINTS

L104-106: This statement could use a reference. 

*** We have added two references to this statement:

Wilson MW, Ridlon AD, Gaynor KM, Gaines SD, Stier AC, Halpern BS. Ecological impacts of human-induced animal behaviour change. Ecol Lett. 2020;23: 1522–1536. doi:https://doi.org/10.1111/ele.13571

Marion S, Demšar U, Davies AL, Stephens PA, Irvine RJ, Long JA. Red deer behavioural response to hiking activity: a study using camera traps. J Zool. 2022;317: 249–261. doi:10.1111/jzo.12976

L132-145: The authors might consider restructuring their set of hypotheses as a numbered list. This could make it easier for the reader (and the authors) to refer back when interpreting the results (e.g., “we found support for Hypothesis 1…”)

*** Thank you for your suggestion, we listed the different hypothesis in the introduction and the relevant places in the discussion. “We hypothesized that (1) the impacts of recreation will differ among mammalian predators and prey, and be influenced by the following factors: (2) trail designation, (3) habitat structure, (4) human footprint, (5) type of management, and (6) season.” L138-140

L179-180: “A systematic random design was used in both areas…” Not clear what the random aspect of the design is. Is this not just systematic sampling?

*** Yes, random was deleted

L362: What are “multivariate response terms”? Do you mean coefficient estimates?

*** Yes modified

I hope you find this useful.

*** Yes, thank you very much for your inputs.

Best,

Justin Suraci

REVIEWER 2

Review for PLOS ONE Mammal responses to human recreation depend to landscape context

I really enjoyed reviewing this manuscript, that I found very interesting, very well written and using an original modelling approach. Results are relevant and will certainly contribute to the growing body of evidence regarding animal responses to outdoor activities and recreation, a potential impact of growing presence and intensity across the globe, with increasing conservation concern. However there is one main point that the authors did not consider that I think might hamper the results presented in the manuscript and several minor comments.

*** Thank you very much for reviewing our manuscript, we are glad we found it interesting, and we have tried to best answer to and mitigate your concerns

My main doubt regards the methodology used to extract the variable describing recreation intensity, namely the Strava variable. Corradini et al. 2021, that you cite, used a different approach to extract an index based on Strava data: they extracted and rasterised the Strava map, rather than using the number of reviews by the users. I am not sure that these two methods are equivalent, does the number of reviews by users really reflect the intensity shown by Strava in their maps? 

*** We understand the confusion about Corradini et al. 2021, that use a different approach. We removed this reference from the Strava section in the method as we used a method adapted from Stelmach and Beddow (2016). We note that our method does not use the reviews of users (as indicated by Reviewer 2 above, this probably refers to the AllTrails data), but rather the count of ‘efforts’ (essentially a unique use of a track). We chose to count efforts rather than users, since some individual users may record many efforts on a single track over time. This method should be more quantitative than the approach used by Corradini et al. 2021 of rasterizing the heatmap; and thus a promising method to capture variable intensities of trail use. As stated by Strava, the heatmap is not quantitative, because the color values of the heatmap are redrawn based on the distribution of the data within a given zoom level (see ‘Heat Normalization’ section of their documentation here: https://medium.com/strava-engineering/the-global-heatmap-now-6x-hotter-23fc01d301de). We added “Our method essentially extracts some of the raw data behind the colorized Strava heatmap, allowing for a quantitative representation”. L297-298

Furthermore, I have an even more serious concern regarding this point: how can you be sure that your index based on Strava data realistically reflects the intensity of recreation? This data certainly needs a validation! Corradini et al. 2021 validated their index using camera-trapping based on trails and found a positive correlation. But this positive correlation needs to be proven locally in every study area one may choose, since it is not guaranteed that Strava data will thoroughly reflect real on-the-ground frequentation of trails. To sum up: 1- I am not sure your method is equivalent to Corradini et al. 2021 one and equally representative; 2 – In any case this index should be validated against another method to make sure it gives a reasonably faithful picture of recreation in your area.

*** See our response to point 1 in the previous comment.

To point 2, we acknowledge your point regarding the validation of the Strava data. As outlined in section 2.3.1, our camera traps were not positioned on recreational trails, preventing us from employing the same validation process as in Corradini et al. 2021. Instead, we opted for an approach that involved utilizing multiple indicators of recreational intensity. This decision was made to encompass the various potential recreational activities that might occur in our study areas and a multiple pathways by which mammals might be perceiving and responding to recreation pressure.

In the method section, we have included the following statement: “We believed these variables encompass various types of recreational activities and proxies for their spatial distribution and intensity in our study areas, in the absence of more direct observations of recreation. By incorporating them in a single analysis, we aimed to provide an holistic representation of recreationists' space use and wildlife perception/response pathways.”L262-266 We hope this addition clarifies our rationale for incorporating diverse indicators to enhance the interpretation of Strava data. 

In the discussion section, we have also expanded on this point by amending/including the following section in the second-to-last paragraph: “Eventually, more research is needed to test and compare different indicators of recreation pressure, including cost-effective indirect measures like those used in our study, and more labour intensive but direct monitoring of recreation activity, such as camera traps. Validation of indirect methods (e.g., Strava use) with direct ground-based monitoring is desirable to better understand their accuracy and limitations in different contexts, and critical before relying on them as a primary monitoring tool. Encouragingly, the more direct monitoring methods are being aided by the incorporation of machine learning tools, for example to process of camera trap images [79]. However, validation will require more purpose-designed sampling. For example, in our study we could not directly validate the Strava data with camera traps since cameras were placed off recreational trails.” L674-683

Other comments:

Lines 82-87: The whole introduction is interesting, well written and well documented. However here I think you lack at least to mention that also hunting is a major factor that can influence avoidance behaviours in mammals. Much research shows that fear of humans can be influenced by the presence and intensity of hunting in a specific area.

*** Yes, we recognize that mentioning hunting is necessary. We added: “Habituation and risk perception can be altered if hunting occurs simultaneously with outdoor recreation, leading to challenges in disentangling the various impacts of human disturbance. [22–24].” L90-91

Lines 104 – 112: Here I would strongly suggest to discuss the potential long-term effects that recreation might have on wildlife species, communities and ecosystem functions. Even when we observe clear avoidance behaviors in mammals towards human recreation we do not know if this avoidance leads to decreased fitness and hence diminishing population trends. One study that evaluated this aspect is Salvatori, M., Oberosler, V., Rinaldi, M., Franceschini, A., Truschi, S., Pedrini, P., & Rovero, F. (2023). Crowded mountains: Long-term effects of human outdoor recreation on a community of wild mammals monitored with systematic camera trapping. Ambio, 52(6), 1085-1097. https://link.springer.com/article/10.1007/s13280-022-01825-w, whose results are relevant also for the interpretation of you results in this manuscript.

*** Thank you for your comment and pointing out this reference. We added: “Long-term exposure to human disturbance can ultimately impact animal fitness (i.e., survival and reproduction success) [34,35].” With the 34: Frid and Dill 2002 and 35: Salvatori et al 2023. L111-112

Additionally, I think that here or in the discussion another point that should be explored is whether recreation can alter herbivory and predation patterns, resulting in direct consequences on other ecosystem components: a relevant subject also for protected area managers. One relevant work here is Di Nicola, W.,Mols, B., & Smit, C. (2023). Human recreation shapes the local scale impact of ungulates on the carbon pools of a temperate coniferous forest. Global Ecology and Conservation, 46, e02574. https://www.sciencedirect.com/science/article/pii/S2351989423002093

*** Thank you for your comments and pointing out this specific paper, we added in the introduction: “Lastly, alterations in herbivory and predation resulting from recreational activities can have broader environmental implications through functional ecosystem change, as demonstrated in Di Nicola et al. 2023, where human-mediated changes were found to impact local carbon stocks”. L112-L115

Lines 183: Here you say that minimum distance from a trail was 50 meters but in Table 1 the minimum distances you report for the 2 areas are well below 50 meters.

*** Thank you for pointing that, we meant 50m from road, not necessary from trail, but we recognized that this statement can be misleading, thus we have deleted this part of the sentence. 

Line 188: I do not understand why you lured your camera trapping sites. The lure effect will have faded a lot before your sampling was ended, since you kept your cameras for more than one year. Luring the site would in this case bias sampling with a higher detection probability at the start of sampling that would have decreased as time passed by. Luring could also introduce biases towards certain species more attracted to the scent, and since you are using community modelling I do not think this is a wise choice.

*** See response to editor above

192: So you controlled cameras only once per year? Didn’t this result in many battery failure and SD card filling?

*** See response to editor above

242: I wonder if one could merge the two types of info, distance from trail and recreation intensity on that trail into a single index weighting both factors together, since animals could remain more distant specifically to those trails more intensely visited by humans. If I am not wrong you did not try the interaction Distance:Strava.

** Thank you for your suggestion; we did initially consider this option. However, we opted for an approach that isolated different aspects of recreation pressure (e.g., distribution, density, intensity) in the hopes of being more informative for management purposes. With our current model framework, practitioners can refer to one metric to reduce mammal – recreationists interactions. However, we agreed than combining pressures and distribution in future studies will be important, thus we added in the discussion: “Likewise, we recommend that future research integrates various measures of recreation into a unified metric to comprehensively grasp how the spatial distribution (e.g., density) and the intensity of recreation (e.g., Strava) collectively influence mammals’ responses.” L683-688

265: This sentence is not clear: what do you mean by entire camera trap sampling area? The area sampled by one single camera trap or the whole study area?

*** We mean the whole study area. We changed this to “until a single cell encompassed the whole study area.” L302-303

Line 274: See my main concern regarding validation of the Strava data.

*** See main answer above where we added: “These complementary variables are believed to encompass various types of recreational activities and their spatial distribution in our study areas. Their combination aims to provide a realistic understanding of recreationists' space use.” 

Line 370: Camera days can also be conveniently shaped as an offset.

*** Yes, however, to our knowledge, and at the time of writing, offset implementation is not possible in a HMSC model. We added some clarification in our Methods: “This is often accounted for using an offset or by creating a detection rate (the number of detections per season divided by the number of active days per season) as a response variable. However, the current version of the HMSC model implementation does not support offset configuration and only supports discrete variables. “L412-416

Line 414: you accidentally put 2 ‘with’ at the end of this line.

***Fixed

Line 437: I would put ‘(see Figure 2)’ to avoid confusion.

***Thanks, done.

Line 488: This last sentence is quite cryptic: why did you used it in combination with non-motorised trails then if it does not reflect vehicle passages at all?

*** This was described in the method section L262-264 of the original manuscript

Line 523: Again, I feel you here overlook the possible effect of hunting in determining avoidance behaviours.

***We added here “Habituation and avoidance behavior can be altered by hunting (Schuttler et al. 2017, Jayakody et al.2008) a factor not accounted for in our study due to a lack of information on spatial variation in hunting pressure. However, previous studies have demonstrated its influence on species responses to recreationists (Jayakody et al. 2008, Kays et al. 2017).” L569-572

Line 532-534: These two possible explanations seem completely in contrast one another.

***Yes, they are contrasting as the two species had two different responses. We changed this to: “Moose and mule deer were exceptions, but with two different responses, with moose using areas with higher trail density more often when there was more forest cover (consistent with the notion that cover provides security), while…” to highlight the difference. L579-582

Line 539: I wonder whether this lack of effect derives from the weakness of the Strava variable you extracted in representing real recreation. Alternatively, it could be that your study areas have relatively low recreation intensity rates, compared to more human dominated landscapes like those of western Europe. Comparing and discussing the responses to recreation between North America and western Europe could also be very interesting.

*** Thank you for your suggestion. Regarding Strava, please see our answer and revisions noted above. It is an interesting idea to compare Europe and North America, while it is not the scope of the paper, we added later in the discussion: “Similarly, these networks can aid in studying interactions between recreation and mammals at a large spatial scale. At an even larger spatial scale, comparing the impacts of recreation on mammals across continents that vary in the intensity and type of recreation activity (e.g. North America vs. western Europe) could potentially enhance our overall understanding of this subject.” L697-701

Line 546: What do you mean with this? What does this difference in home range size entail?

*** This was confusing, we edited this section to now say: “However, in the case of red squirrels, the lack of avoidance might indicate an inability to avoid disturbance rather than habituation to stress and recreation due to their small home range.” L595-597

Lines 555 – 557: I did not understand what you mean here with leveraging, please clarify.

***We recognized that this sentence was not clear nor very informative and removed it

Line 622: ‘tpressure’ typo.

***Fixed

Line 658: ‘ewithin’ typo.

***Fixed

Line 664: Here you wisely mention the potential long-term effect on animal population demography potentially driven by avoidance of human recreation: see the paper I cited above that tackled this topic.

***Thank you for your comment, we added “.. to understand long term effects of human disturbances [34,35]” with the two references Frid and Dill 2002 and 35: Salvatori et al 2023 L733-734.

---

## [Decision Letter · Decision Letter 1]

9 Feb 2024

PONE-D-23-21203R1Mammal responses to human recreation depend on landscape contextPLOS ONE

Dear Dr. Marion,

Thank you for submitting your manuscript to PLOS ONE. After careful consideration, we feel that it has merit but does not fully meet PLOS ONE’s publication criteria as it currently stands. Therefore, we invite you to submit a minor revision of the manuscript that addresses the points raised during the review process.

Thank you for adequately incorporating the reviewers suggestions. There are a few small pending comments from Reviewer 2 at the bottom of this message that should be quick to address.

We look forward to receiving your revised manuscript.

Kind regards,

Bogdan Cristescu

Academic Editor

PLOS ONE

Journal Requirements:

Reviewers' comments:

Reviewer's Responses to Questions

**Comments to the Author**

1. If the authors have adequately addressed your comments raised in a previous round of review and you feel that this manuscript is now acceptable for publication, you may indicate that here to bypass the “Comments to the Author” section, enter your conflict of interest statement in the “Confidential to Editor” section, and submit your "Accept" recommendation.

Reviewer #1: All comments have been addressed

Reviewer #2: All comments have been addressed

2. Is the manuscript technically sound, and do the data support the conclusions?

Reviewer #1: Yes

Reviewer #2: Yes

3. Has the statistical analysis been performed appropriately and rigorously? 

Reviewer #1: Yes

Reviewer #2: Yes

4. Have the authors made all data underlying the findings in their manuscript fully available?

Reviewer #1: (No Response)

Reviewer #2: Yes

5. Is the manuscript presented in an intelligible fashion and written in standard English?

Reviewer #1: Yes

Reviewer #2: Yes

6. Review Comments to the Author

Reviewer #1: (No Response)

Reviewer #2: Second round of review for PLOS ONE Mammal responses to human recreation depend on landscape context.

I thank the author for revising their manuscript and for considering my comments. I think they addressed most of the points raised with sufficient detail. I think there are only a couple of points that would require further clarifications:

Regarding the STRAVA variable, I still do not understand how you extracted the data and how these data relate to the heatmap. Were your data all those on which STRAVA creates the heatmap or only a part of it? Aren’t the segments linked to certain specific trails on which runners or bikers compete remotely through the app or do they have the same spatial representation of the heatmap? Are these data freely available or you had to purchase them? Since you say that your data are only some of those used for the heatmap I wonder if they are enough to get a thorough picture of outdoor recreation in your area.

I think that the sentence “Long-term exposure to human disturbance can ultimately impact animal fitness (i.e., survival and reproduction success)” that you added is misleading. Exposure to human disturbance can for sure lead to decreased fitness but this is not a necessary outcome, indeed Salvatori et al. (2023) found that even under intense human frequentation and in presence of marked spatio-temporal avoidance behaviour mammal occupancy can increase both at species and community level within a protected area in Europe. I think that a little bit more discussion about the possible outcomes of human disturbance in the long term in light of this and other research could help understand the potential consequences for animal populations, which is why we do this kind of studies in the first place.

7. PLOS authors have the option to publish the peer review history of their article (what does this mean?). If published, this will include your full peer review and any attached files.

Reviewer #1: **Yes: **Justin Suraci

Reviewer #2: No

---

## [Author Response · Author response to Decision Letter 1]

26 Feb 2024

Reviewer #1: (No Response)

Reviewer #2: Second round of review for PLOS ONE Mammal responses to human recreation depend on landscape context.

I thank the author for revising their manuscript and for considering my comments. I think they addressed most of the points raised with sufficient detail. I think there are only a couple of points that would require further clarifications:

*** Thank you for reviewing our paper a second time and for your comments

Regarding the STRAVA variable, I still do not understand how you extracted the data and how these data relate to the heatmap. 

Were your data all those on which STRAVA creates the heatmap or only a part of it? 

***Thanks for your comment. It is difficult to say exactly how the extracted data relate to the STRAVA heatmap, as STRAVA provides very limited information on how the data are quantitatively converted to the heatmap, and it appears that they have been updating and adapting their process over time. STRAVA does state that their heatmap is just a snapshot of historical data, but it is not clear if is all available historical data. They also don’t say what data are used to create the heatmap (e.g., Is the total number individual athletes or efforts, which can count a single athlete multiple times? Do they normalize data to account for the fact that some segments have been around longer than others? Etc.) Additionally, they do not provide any details on how they convert numbers to colors, but they do mention that that segments with very little activity may not show up at all, which suggests they set the minimum value for their colorization to a value greater than 1 (more info here: https://support.strava.com/hc/en-us/articles/216918877-Strava-Metro-and-the-Global-Heatmap).

All we can say is that the data we are able to extract through their API is certainly used in the heatmap. We mentioned the heatmap only because we expect readers are more likely to be familiar with the heatmap than the API, and we thought it would help contextualize what kind of data we are pulling through their API. We recognize that our sentence “Our method essentially extracts some of the raw data behind the colorized Strava heatmap, allowing for a quantitative representation” might have been confusing. We have expanded it to read: “For those familiar with the Strava Global Heat Map (https://support.strava.com/hc/en-us/articles/216918877-Strava-Metro-and-the-Global-Heatmap), we note that while our methodology pulls raw data used in the heatmap, it may generate different results for a variety of reasons (e.g., the heatmap is updated monthly, low activity segments do not show up on the heatmap, some of the data in the heatmap may not be publicly available, etc.). However, since the quantitative values of the heatmap colors are not available, and the colorization technique is poorly described, we chose this approach as a reproducible method to generate quantitative data.”

 Aren’t the segments linked to certain specific trails on which runners or bikers compete remotely through the app or do they have the same spatial representation of the heatmap? 

*** The segments should have the same spatial representation as the heatmap (as far as we know, the heatmap shows individual segments). Strava data are stored in segments, not trails. Segments are created by users, and they are not necessarily associated with any outside data about existing trail networks. When users track an activity, they can choose to track themselves as a brand new segment that they create, or along an existing segment someone else created. We also note that each segment is associated with a specific activity. This means that runner and biker data can be analyzed separately or together. In our case, we have combined the data from runners and bikers’ activities. Additionally, we have added to the text: ‘Each segment can be associated with a specific activity that can be analyzed separately or, as in our case, together.

Are these data freely available or you had to purchase them? 

*** Freely available. We have edited L299-300 to be clearer about this to read: “We extracted Strava data using their Python Application Programming Interface (API), which is freely available to anyone with a Strava account.”. 

Since you say that your data are only some of those used for the heatmap I wonder if they are enough to get a thorough picture of outdoor recreation in your area.

*** Thank you for your comment and we appreciate this point. Firstly, please see our response to the comment above that now better describes what we know about the relationship between the data we extracted and the heatmap. Secondly, since there are currently no data sources available to give a thorough picture of spatial patterns of outdoor recreation in our study area, our approach was to use multiple variables that encompass various types of recreational activities and provide a more holistic representation of recreactionists’ space use, as mentioned L266-268. Thus, we agree that Strava as a single variable might not represent all recreation in the area. However, there are few other free and accessible methods to get such fine-scaled (trail/segment) relative use across large areas, thus we feel it is warranted as a ‘best available’ dataset to bring into the study. The combination with the other variable aims to balance out each data source’s limitations.

Since there are myriad ways in which Strava (or any other similar data source) may be limited, all of which are likely highly context dependent, we chose to be concise with our discussion of its limitations in the Discussion (e.g., see L688-690: “Validation of indirect methods (e.g., Strava use) with direct ground-based monitoring is desirable to better understand their accuracy and limitations in different contexts, and critical before relying on them as a primary monitoring tool.”). Furthermore, we feel that the general approach of combining recreation data sources and exploring their relationship to wildlife using the framework of this study was the primary objective of this manuscript, rather than assessing the value of any single data source (e.g., see L694-697: “Likewise, we recommend that future research integrates various measures of recreation into a unified metric to comprehensively grasp how the spatial distribution (e.g., density) and the intensity of recreation (e.g., Strava) collectively influence mammals’ responses”).

I think that the sentence “Long-term exposure to human disturbance can ultimately impact animal fitness (i.e., survival and reproduction success)” that you added is misleading. Exposure to human disturbance can for sure lead to decreased fitness but this is not a necessary outcome, indeed Salvatori et al. (2023) found that even under intense human frequentation and in presence of marked spatio-temporal avoidance behaviour mammal occupancy can increase both at species and community level within a protected area in Europe. I think that a little bit more discussion about the possible outcomes of human disturbance in the long term in light of this and other research could help understand the potential consequences for animal populations, which is why we do this kind of studies in the first place.

*** Thank you for your comment on this specific point. We have removed the 'ultimately' from the sentence. Additionally, using your example, we have added the following:

“However, different outcomes of human disturbance impacting the long-term can be found. Salvatori et al. (2023) [37] found that both community and species-level occurrences increased within a protected area in Europe, despite human activities causing strong temporal avoidance in the whole community. This highlights the complexity of human-wildlife interaction and the necessity of long-term studies.”

---

## [Editor Report · Decision Letter 2]

6 Mar 2024

Mammal responses to human recreation depend on landscape context

PONE-D-23-21203R2

Dear Dr. Marion,

We’re pleased to inform you that your manuscript has been judged scientifically suitable for publication and will be formally accepted for publication once it meets all outstanding technical requirements.

Congratulations on your paper.

Kind regards,

Bogdan Cristescu

Academic Editor

PLOS ONE

---

## [Editor Report · Acceptance letter]

22 Mar 2024

PONE-D-23-21203R2 

PLOS ONE

Dear Dr. Marion, 

I'm pleased to inform you that your manuscript has been deemed suitable for publication in PLOS ONE. Congratulations! Your manuscript is now being handed over to our production team.

Kind regards, 

on behalf of

Dr. Bogdan Cristescu 

Academic Editor

PLOS ONE